# Informed Machine Learning with a Stochastic-Gradient-based Algorithm for Training with Hard Constraints

## Abstract

A methodology for informed machine learning is presented and its effectiveness is shown through numerical experiments with physics-informed learning problems. The methodology has three main distinguishing features. Firstly, prior information is introduced in the training problem through hard constraints rather than through the typical modern practice of using soft constraints (i.e., regularization terms). Secondly, the methodology does not employ penalty-based (e.g., augmented Lagrangian) methods since the use of such methods results in an overall methodology that is similar to a soft-constrained approach. Rather, the methodology is based on a recently proposed stochastic-gradient-based algorithm that maintains computationally efficiency while handling constraints with a Newton-based technique. Thirdly, a new projection-based variant of the well-known Adam optimization methodology is proposed for settings with hard constraints. Numerical experiments on a set of physics-informed learning problems show that, when compared with a soft-constraint approach, the proposed methodology can be easier to tune, lead to accurate predictions more quickly, and lead to better final prediction accuracy.

## 1 Introduction

In this paper, we propose a methodology for informed supervised machine learning and demonstrate its effectiveness on a set of challenging test problems. The methodology involves incorporating prior knowledge into the learning process through *hard constraints* that are imposed *during training only*. Both of these highlighted aspects are critical for its effectiveness. Our methodology's use of *hard constraints* is in contrast to previously proposed methodologies that incorporate prior knowledge through either (a) soft constraints (Zhu et al., 2019) (i.e., through regularization/penalty terms in the objective function) or (b) designing the prediction function to incorporate knowledge directly (Chalapathi et al., 2024; Négiar et al., 2023) (e.g., through neural network layers for which a forward pass requires solving a set of equations or even an optimization problem). By imposing such constraints *during training only*, one can avoid having the trained network require expensive operations for each forward pass. Another key feature of our proposed methodology is that we do not solve the hard-constrained training problem with a penalty-based (e.g., augmented Lagrangian) method. This feature is also critical for the effectiveness of our methodology. Ours is not the first article to propose the use of hard constraints for informed learning; see, e.g., Márquez-Neila et al. (2017). However, some other approaches for training with hard constraints that have been proposed use such penalty-based methods, even though this ultimately means that the behavior of the training algorithm is similar to employing an unconstrained training algorithm to a soft-constrained formulation; see, e.g., Lu et al. (2021). Our intuition is that hard constraints guide the neural network to prioritize mapping the PDE solution, e.g. in physics-informed machine learning, even only at certain inputs, enabling faster and more efficient training. For our methodology to solve the hard-constrained problems, we propose a variant of the stochastic-gradient-based algorithm from Berahas et al. (2021) that handles the constraints with a Newton-based technique; see also Berahas et al. (2023). An additional important feature of our proposed methodology is a new projection-based variant of the well-known Adam (Kingma & Ba, 2015) optimization routine. (Our framework is easily extended to other diagonal-scaling methods, such as Adagrad (Duchi et al., 2011) and RMSprop (Dauphin et al., 2015; Tieleman & Hinton, 2012). We simply demonstrate our framework using Adam

due to its popularity in the supervised learning literature (Rathore et al., 2024).) By decomposing the stochastic algorithm's step computation routine and exploiting the fact that the Adam scaling matrix is diagonal and positive definite, the per-iteration cost of our algorithm is comparable to that of an Adam-based method employed in a soft-constraint/regularization approach.

As a case study for demonstrating the effectiveness of our proposed methodology, we consider a set of physics-informed learning problems. Through straightforward formulations of such problems, we show that our stochastic algorithmic framework with projected Adam (P-Adam) scaling outperforms alternative approaches involving only soft constraints or a non-projection-based Adam routine. In particular, we show that our approach can (a) yield good solutions more quickly, (b) yield more accurate predictions after training is completed, and (c) require less hyperparameter tuning.

## 1.1 CONTRIBUTIONS AND LIMITATIONS

We propose a methodology for informed supervised machine learning that (a) incorporates prior knowledge through hard constraints, (b) employs a computationally efficient method that handles the objective with a stochastic-gradient-based technique and handles the constraints with a Newton-based technique such that the overall per-iteration cost is comparable to that of alternative soft-constrained approaches, and (c) involves a new projection-based variant of Adam. We show that our method yields superior performance on a test set of physics-informed learning problems.

Our proposed algorithm is based on a method that enjoys state-of-the-art convergence and complexity guarantees. We do not show that these guarantees extend to the setting when our P-Adam routine is employed, although our experiments show that the routine is robust. In addition, our approach has a higher per-iteration cost when compared to a method that may be employed to minimize a soft-constraint-based (unconstrained) training problem. However, our experiments show that, within the same computational time budget, our approach can offer better trained models. Also, the fact that—through our methodology—one might avoid having to employ a model that requires expensive operations with every forward pass can justify this additional per-iteration cost during model training.

## 2 STOCHASTIC-GRADIENT-BASED, HARD-CONSTRAINED TRAINING

The supervised training of a machine learning model involves solving an optimization problem over a set of parameters of a prediction function, call it $p : \mathbb{R}^{n_f} \times \mathbb{R}^d \to \mathbb{R}^{n_o}$, where $n_f$ is the number of features in an input vector, $d$ is the dimension of the training/optimization problem, and $n_o$ is the dimension of the output vector. Denoting known input-output pairs in the form $(x, y) \in \mathbb{R}^{n_f} \times \mathbb{R}^{n_o}$ and given a loss function $\ell : \mathbb{R}^{n_o} \times \mathbb{R}^{n_o} \to \mathbb{R}$, the training/optimization problem can be viewed in expected-loss or empirical-loss minimization form, i.e., respectively,

$$\min_{w \in \mathbb{R}^d} \int_{\mathcal{X} \times \mathcal{Y}} \ell(p(x, w), y) \mathrm{d}\mathbb{P}(x, y) \quad \approx \quad \min_{w \in \mathbb{R}^d} \frac{1}{N} \sum_{i=1}^{N} \ell(p(x_i, w), y_i),$$

where $\mathcal{X}$ is the input domain, $\mathcal{Y}$ is the output domain, $\mathbb{P}$ is the input-output probability function, and $\{(x_i, y_i)\}_{i=1}^{N} \subset \mathbb{R}^{n_f} \times \mathbb{R}^{n_o}$. In our setting, the problem has (hard) constraints on $w$ as well. Generally, these can be formulated in various ways; e.g., expectation, probabilistic, or almost-sure constraints. We contend that for many informed-learning problems—such as for many physics-informed learning problems, as we discuss in §3—a fixed, small number of constraints suffices to improve training. Given a (small) number $m$ of input-output pairs $\{(x_i^c, y_i^c)\}_{i=1}^{m}$, the constraints may take the form

$$\phi_i(p(x_i^c, w), y_i^c, \dots) = 0 \ \text{ for all } \ i \in \{1, \dots, m\},$$

where the arguments to the constraint functions $\{\phi_i\}$ may include additional terms, such as derivatives of the prediction function with respect to inputs and/or model weights; see §4 for specific examples.

### 2.1 STOCHASTIC-GRADIENT-BASED METHOD THAT HANDLES CONSTRAINTS WITH A NEWTON-BASED TECHNIQUE

For the sake of notational simplicity, let us proceed in this section with the problem formulation

$$\min_{w \in \mathbb{R}^d} f(w) \ \text{ subject to } \ c(w) = 0 \ \text{ with } \ f(w) = \mathbb{E}[F(w, \omega)], \tag{1}$$

where $f : \mathbb{R}^d \to \mathbb{R}$, $c : \mathbb{R}^d \to \mathbb{R}^m$, $\omega$ is a random variable with associated probability space $(\Omega, \mathcal{F}, \mathbb{P})$, $F : \mathbb{R}^d \times \Omega \to \mathbb{R}$ and $\mathbb{E}[\cdot]$ denotes expectation with respect to $\mathbb{P}$. The algorithm that we propose follows the stochastic-gradient-based Sequential Quadratic Programming (SQP) framework proposed and analyzed in Berahas et al. (2021) for solving constrained optimization problems. We state a simplified version of the method as Algorithm 1 below. The algorithm requires Lipschitz constants for the objective gradient and constraint Jacobian, which are denoted as $L_{\nabla f} \in \mathbb{R}_{>0}$ and $L_{\nabla c} \in \mathbb{R}_{>0}$, respectively. In practice these values can be estimated, e.g., as in Berahas et al. (2021). We emphasize that when the Jacobian $\nabla c(w_k)^T$ has a small number of rows and $H_k$ is a positive definite diagonal matrix and for all $k \in \mathbb{N}$, the linear system (2) can be computed with a computational cost that is proportional to that of computing $H_k^{-1} g_k$, as shown later in this section.

---

**Algorithm 1** Stochastic-Gradient-based SQP Framework (Berahas et al., 2021)

---

**Require:** $w_1 \in \mathbb{R}^d$, $(L_{\nabla f}, L_{\nabla c}) \in \mathbb{R}_{>0} \times \mathbb{R}_{>0}$, $(\tau, \xi) \in \mathbb{R}_{>0} \times \mathbb{R}_{>0}$, and $\{\bar{\alpha}_k\} \subset (0, 1]$
1: **for** all $k \in \mathbb{N}$ **do**
2:      Compute a stochastic gradient estimate $g_k \approx \nabla f(w_k)$ and choose symmetric $H_k \in \mathbb{R}^{d \times d}$
3:      Compute $s_k$ by solving

$$\begin{bmatrix} H_k & \nabla c(w_k) \\ \nabla c(w_k)^T & 0 \end{bmatrix} \begin{bmatrix} s_k \\ \lambda_k \end{bmatrix} = - \begin{bmatrix} g_k \\ c(w_k) \end{bmatrix} \qquad (2)$$

4:      Set $w_{k+1} \leftarrow w_k + \alpha_k s_k$, where $\alpha_k \leftarrow (\bar{\alpha}_k \xi \tau)/(\tau L_{\nabla f} + L_{\nabla c})$
5: **end for**

---

For the sake of completeness, we state the following theorem pertaining to convergence guarantees of Algorithm 1 for solving (1), for which $(w, \lambda) \in \mathbb{R}^d \times \mathbb{R}^m$ satisfies first-order stationarity conditions if and only if $\nabla f(w) + \nabla c(w)\lambda = 0$ and $c(w) = 0$. The theorem shows that the algorithm guarantees asymptotic convergence in probability of the sequence of primal iterates to stationarity, which for the sake of analysis is described in terms of the stochastic process $\{W_k\}$, which in turn are determined by the process of stochastic gradient estimators $\{G_k\}$ that generate a sequence of sub-$\sigma$-algebras $\{\mathcal{F}_k\}$. The theorem refers to "true" quantities that would be computed with exact objective-gradient information even though Algorithm 1 only uses stochastic-gradient estimates.

**Theorem 1** (see Berahas et al. (2021, Corollary 3.14) and Curtis et al. (2023a, Equation (16))). *Suppose there exists convex $\mathcal{W} \subseteq \mathbb{R}^d$ containing the stochastic process $\{W_k\}$ generated by Algorithm 1 almost-surely such that $f : \mathbb{R}^d \to \mathbb{R}$ is continuously differentiable and bounded over $\mathcal{W}$, $\nabla f : \mathbb{R}^d \to \mathbb{R}^d$ is bounded and Lipschitz continuous over $\mathcal{W}$, $c : \mathbb{R}^d \to \mathbb{R}^m$ is Lipschitz continuous, continuously differentiable, and bounded over $\mathcal{W}$, and $\nabla c^T : \mathbb{R}^d \to \mathbb{R}^{m \times d}$ is Lipschitz continuous with singular values bounded uniformly away from zero over $\mathcal{W}$. In addition, suppose that there exists a tuple $(\sigma, \zeta, \kappa) \in \mathbb{R}_{>0} \times \mathbb{R}_{>0} \times \mathbb{R}_{>0}$ such that, for all $k \in \mathbb{N}$, one has $\mathbb{E}[G_k | \mathcal{F}_k] = \nabla f(W_k)$, $\mathbb{E}[\|G_k - \nabla f(W_k)\|_2^2 | \mathcal{F}_k] \leq \sigma$, and $H_k$ is $\mathcal{F}_k$-measurable with $\|H_k\|_2 \leq \kappa$ and $u^T H_k u \geq \zeta \|u\|_2^2$ for all $u \in \mathrm{Null}(\nabla c(W_k)^T)$. Finally, suppose that $(\tau, \xi) \in \mathbb{R}_{>0} \times \mathbb{R}_{>0}$ is chosen such that, for all $k \in \mathbb{N}$, one finds $\xi \leq \bar{\xi}_k$, where*

$$\bar{\xi}_k := \begin{cases} \infty & \text{if } S_k = 0 \\ \frac{-\tau(G_k^T S_k + \frac{1}{2} S_k^T H_k S_k) + \|c(w_k)\|_1}{\tau \|S_k\|_2^2} & \text{otherwise,} \end{cases}$$

*and $\tau \leq \bar{\tau}_k^{true}$, where, with $S_k^{true}$ being the first component of the solution of (2) that would be computed if $G_k$ were replaced by $\nabla f(W_k)$, one defines*

$$\bar{\tau}_k^{true} := \begin{cases} \infty & \text{if } \nabla f(W_k)^T S_k^{true} + (S_k^{true})^T H_k S_k^{true} \leq 0 \\ \frac{\frac{1}{2}\|c(W_k)\|_1}{\nabla f(W_k)^T S_k^{true} + (S_k^{true})^T H_k S_k^{true}} & \text{otherwise.} \end{cases}$$

*Then, if $\{\bar{\alpha}_k\}$ is monotonically nonincreasing ($\bar{\alpha}_{k+1} \leq \bar{\alpha}_k$ for all $k \in \mathbb{N}$), unsummable ($\sum_{k=1}^{\infty} \bar{\alpha}_k = \infty$), and square-summable ($\sum_{k=1}^{\infty} \bar{\alpha}_k^2 < \infty$) with $\bar{\alpha}_1 \in \mathbb{R}_{>0}$ sufficiently small, one has that*

$$\liminf_{k \to \infty} \mathbb{E}[\|\nabla f(W_k) + \nabla c(W_k)\Lambda_k^{true}\|_2^2 + \|c(W_k)\|_2] = 0,$$

*where for all $k \in \mathbb{N}$ the vector $\Lambda_k^{true}$ is the latter component of the solution of (2) that would be computed if $G_k$ were replaced by $\nabla f(W_k)$. That is, under these conditions, the sequence $\{W_k\}$ corresponds to a sequence of first-order stationarity measures that vanishes.*

Under these conditions and additional assumptions (see Curtis et al. (2023a) for details), the sequence of iterates $\{W_k\}$ generated by Algorithm 1 converges almost-surely to a primal stationary point and a running average of the sequence $\{\Lambda_k\}$ generated by Algorithm 1 converges almost-surely to a dual stationary point. In addition, in Curtis et al. (2023c), it is shown that the algorithm enjoys worst-case complexity guarantees that are on par with those of stochastic-gradient-based methods for unconstrained optimization. The rate of convergence of the constraint violation is improved by a two-step size SQP method in O'Neill (2024).

## 2.2 A NEW PROJECTION-BASED VARIANT OF ADAM

The practical performance of stochastic-gradient-based methods for (unconstrained) training can be improved significantly with adaptive scaling. Approaches of this type include Adagrad (Duchi et al., 2011), RMSProp (Dauphin et al., 2015; Tieleman & Hinton, 2012), and Adam (Kingma & Ba, 2015), the popular variants of which involve only diagonal scaling matrices. One approach for employing, say, an Adam-based scheme in the context of a Newton-based algorithm for constrained optimization is to choose the matrix $H_k$ in (2) as such a positive definite diagonal scaling matrix and to replace the first component on the right-hand side of (2) with the running average of gradients that is employed in Adam for unconstrained training/optimization. Such an approach was proposed and tested in Márquez-Neila et al. (2017).

We propose a new projection-based variant of Adam for constrained optimization, presented in Algorithm 2. It is similar to that proposed in Márquez-Neila et al. (2017), but takes into account the fact that, in the setting of a Newton-based method for constrained optimization, *the component of a stochastic-gradient estimate that lies in the range space of the constraint derivative does not affect the search direction*. Therefore, the idea proposed here projects-out this component for the running averages of gradient values. In other words, our Algorithm2 mainly differs from Márquez-Neila et al. (2017) in that Algorithm2 utilizes the momentum of the component of the stochastic gradient in the null space of $\nabla c(w_k)^T$ *rather than the entire stochastic gradient*, which is the case in Márquez-Neila et al. (2017). This is a nontrivial change. In our experiments, this distinction demonstrates the superior performance of our method compared to that of Márquez-Neila et al. (2017).

A Newton-based step for the constraints and a steepest-descent-type step for the objective can be computed at $w_k$ by solving (2) with $H_k = I$. By the fundamental theorem of linear algebra, let $s_k = v_k + u_k$, where $v_k \in \text{Range}(\nabla c(w_k))$ and $u_k \in \text{Null}(\nabla c(w_k)^T)$. The second row of (2) implies $v_k = -\nabla c(w_k)(\nabla c(w_k)^T \nabla c(w_k))^{-1}c(w_k)$, which is unaffected by $g_k$. On the other hand, with the columns of an orthogonal matrix $Z_k$ spanning the null space of $\nabla c(w_k)^T$, one finds $u_k = -Z_k(Z_k^T Z_k)^{-1}Z_k^T g_k$ is the orthogonal projection of $g_k$ onto $\text{Null}(\nabla c(w_k)^T)$.

The discussion in the prior paragraph shows that the solution component $s_k$ in (2) with $H_k$ replaced by $I$ is the same when the stochastic gradient $g_k$ is replaced by its orthogonal projection onto the null space of $\nabla c(w_k)^T$, which is to say that this projection of $g_k$ is what matters for the search direction component. This suggests that an Adam-based approach can be employed where the scaling matrix and right-hand side vectors are computed based on the projection of $g_k$, rather than on $g_k$ itself. Since it is not computationally tractable to compute a null-space basis matrix $Z_k$ for all $k \in \mathbb{N}$, one can instead employ the projection operator as shown in Algorithm 2 below.

Algorithm 2 offers various alternatives as well. For example, a projection-based variant of Adagrad is obtained when, instead, $\widehat{p}_k \leftarrow g_k$ and $\widehat{q}_k \leftarrow \widehat{q}_{k-1} + (g_k \circ g_k)$. The major computation cost of Algorithm 2 per iteration is from Line 3 and Line 8. For Line 3, the cost of $(\nabla c(w_k)^T \nabla c(w_k))^{-1}$ is $\mathcal{O}(m^2 d + m^3)$. Hence the cost of computing $\bar{g}_k$ is $\mathcal{O}(m^2 d + m^3)$. The cost of computing $s_k$ in Line 8 is also $\mathcal{O}(m^2 d + m^3)$, as shown next. Therefore, the overall cost of Algorithm 2 is $\mathcal{O}(m^2 d + m^3)$. When the number of rows of $\nabla c(w_k)^T$ (i.e., $m$) is small, the overall cost per iteration is proportional to that of computing $H^{-1}g$ with a diagonal and positive definite $H$, as is required for an Adam-based method for the unconstrained setting.

## 2.3 SOLVING THE LINEAR SYSTEMS

In each iteration, Algorithm 1 (with Algorithm 2) requires solving a linear system of the form

$$\begin{bmatrix} H & J^T \\ J & 0 \end{bmatrix} \begin{bmatrix} s \\ \lambda \end{bmatrix} = - \begin{bmatrix} p \\ c \end{bmatrix}, \tag{3}$$

---

**Algorithm 2** P-Adam, Projection-based Adam

---

**Require:** $w_1 \in \mathbb{R}^d$, $(L_{\nabla f}, L_{\nabla c}) \in \mathbb{R}_{>0} \times \mathbb{R}_{>0}$, $(\tau, \xi) \in \mathbb{R}_{>0} \times \mathbb{R}_{>0}$, $p_0$ and $q_0$ are zero vectors, $\beta_1 \in (0,1)$, $\beta_2 \in (0,1)$, $\mu \in \mathbb{R}_{>0}$, and $\{\bar{\alpha}_k\} \subset (0,1]$

1: **for** all $k \in \mathbb{N}$ **do**
2:     Compute a stochastic gradient estimate $g_k \approx \nabla f(w_k)$
3:     Compute $\bar{g}_k \leftarrow (I - \nabla c(w_k)(\nabla c(w_k)^T \nabla c(w_k))^{-1} \nabla c(w_k)^T) g_k$
4:     Set $p_k \leftarrow \beta_1 p_{k-1} + (1 - \beta_1)\bar{g}_k$
5:     Set $q_k \leftarrow \beta_2 q_{k-1} + (1 - \beta_2)(\bar{g}_k \circ \bar{g}_k)$, where $(\bar{g}_k \circ \bar{g}_k)_i = (\bar{g}_k)_i^2$ for all $i \in \{1, \ldots, d\}$
6:     Set $\widehat{p}_k \leftarrow (1/(1 - \beta_1^k))p_k$
7:     Set $\widehat{q}_k \leftarrow (1/(1 - \beta_2^k))q_k$
8:     Compute $s_k$ by solving $\begin{bmatrix} \mathrm{diag}(\sqrt{\widehat{q}_k + \mu}) & \nabla c(w_k) \\ \nabla c(w_k)^T & 0 \end{bmatrix} \begin{bmatrix} s_k \\ \lambda_k \end{bmatrix} = - \begin{bmatrix} \widehat{p}_k \\ c(w_k) \end{bmatrix}$
9:     Set $w_{k+1} \leftarrow w_k + \alpha_k s_k$, where $\alpha_k \leftarrow (\bar{\alpha}_k \xi \tau)/(\tau L_{\nabla f} + L_{\nabla c})$
10: **end for**

---

where $H \in \mathbb{R}^{d \times d}$ is diagonal and positive definite and $J \in \mathbb{R}^{m \times d}$ has full row rank. Our focus is on computing $s \in \mathbb{R}^d$. When $m \ll d$, as in the context of the problems considered in this paper, this solution component can be computed with relatively low computational cost through a decomposition.

Let $s = v + u$ with $v \in \mathrm{Range}(J^T)$ and $u \in \mathrm{Null}(J)$. As mentioned, (3) gives $v = -J^T(JJ^T)^{-1}c$. Thus, $v$ can be computed by solving an $m$-dimensional positive definite system $JJ^T\tilde{v} = -c$ for $\tilde{v} \in \mathbb{R}^m$, then computing $J^T\tilde{v} = v$. Now letting $Z \in \mathbb{R}^{d \times (d-m)}$ denote an orthogonal matrix whose columns span $\mathrm{Null}(J)$, the first row of (3) states $Hs + J^T\lambda = -(p + Hv)$, so $u = -Z(Z^THZ)^{-1}Z^TH(H^{-1}p+v)$. However, this is not efficient since it requires $Z$. Fortunately, one can replace $Z(Z^THZ)^{-1}Z^TH$ with a matrix in terms of $H$ and $J$, as we now explain.

The $H$-inner-product is defined by $\langle a,b \rangle_H = a^THb$ for all $(a,b) \in \mathbb{R}^d \times \mathbb{R}^d$. Given a matrix $P \in \mathbb{R}^{d \times d}$, its $H$-adjoint is the matrix $P^* \in \mathbb{R}^{d \times d}$ such that $\langle a, Pb \rangle_H = \langle P^*a, b \rangle_H$ for all $(a,b) \in \mathbb{R}^d \times \mathbb{R}^d$. Since $H \succ 0$, it can be verified easily that $P^* = H^{-1}P^TH$. One calls $P$ an $H$-orthogonal-projection matrix if and only if it is idempotent (i.e., $P = P^2$) and $H$-self-adjoint (i.e., $P = P^*$). Moreover, $P$ projects onto $\mathrm{span}(Z)$ if and only if $Pa = Zb$ for some $b$ for all $a \in \mathbb{R}^d$.

One finds $Z(Z^THZ)^{-1}Z^TH = I - H^{-1}J^T(JH^{-1}J^T)^{-1}J$ is the unique $H$-orthogonal-projection operator onto $\mathrm{Null}(J)$, so $u = -(I - H^{-1}J^T(JH^{-1}J^T)^{-1}J)(H^{-1}p+v)$. Thus, $u$ can be computed by: scaling $J$ and $p$ to form $\tilde{J}^T := H^{-1}J^T$ and $\tilde{p} := H^{-1}p$; computing $\tilde{H} := J\tilde{J}^T$ and $\hat{p} := J(\tilde{p} + v)$; solving an $m$-dimensional positive definite system $\tilde{H}r = \hat{p}$ for $\hat{p} \in \mathbb{R}^m$; multiplying $\tilde{J}^T\hat{p}$; and computing a few sums. Similar to $\bar{g}_k$, the cost of computing $v$ and $u$ are $\mathcal{O}(m^2d + m^3)$. Therefore, since $s = v + u$, the cost of computing $s$ is also $\mathcal{O}(m^2d + m^3)$.

## 3 PHYSICS-INFORMED LEARNING PROBLEMS

Our stochastic-gradient-based algorithm with P-Adam scaling for solving hard-constrained problems can be employed in numerous informed-learning contexts (e.g., fair learning (Curtis et al., 2023b; Donini et al., 2018; Komiyama et al., 2018; Zafar et al., 2017a;b; 2019)). For this work, we tested our approach on a few physics-informed learning problems. We emphasize that our goal here is not to test huge-scale, state-of-the-art techniques for physics-informed learning. Rather, we take a few physics-informed learning test problems and train relatively straightforward neural networks in order to demonstrate the relative performance of our proposed algorithm with a soft-constrained approach with Adam scaling (Kingma & Ba, 2015) and a hard-constrained approach with Adam (not P-Adam) scaling (Márquez-Neila et al., 2017). The relative performance of the algorithms would be similar if we were to train much more sophisticated and large-scale neural networks that are being developed in state-of-the-art physics-informed learning. For more on physics-informed learning we direct the reader to, e.g., Cuomo et al. (2022); Karniadakis et al. (2021); Lagaris et al. (1998); Raissi et al. (2019); Takamoto et al. (2023); Wang et al. (2021; 2023). The work (Chen et al., 2024) lies in the physics-informed learning with hard constraints, but is restrict to the hard constraints that the PDE inputs and solutions are linearly related, whereas our method handles general nonlinear constraints.

They enforce feasibility via projection, while we allow infeasible iterates, using projection only for momentum. Thus, we do not compare our method with theirs.

Let us now provide an overview of the setting of physics-informed learning that we consider in our experiments. A parametric partial differential equation (PDE) can be written generically as $\mathcal{F}(\phi, u) = 0$, where $(\Phi, \mathcal{U}, \mathcal{V})$ is a triplet of Banach spaces, $\mathcal{F} : \Phi \times \mathcal{U} \to \mathcal{V}$ is a differential operator, $\phi \in \Phi$ represents PDE parameters, and $u \in \mathcal{U}$ denotes a solution of the PDE corresponding to $\phi$. The aim is to train a model to learn a mapping from the PDE parameters to a corresponding solution. Let (an approximation of) such a mapping be denoted as $\mathcal{G} : \Phi \times \mathbb{R}^n \times \mathbb{R}^d \to \mathcal{U}$, the inputs to which are PDE parameters, a vector encoding information about the domain of the PDE solution about which one aims to make a prediction (e.g., temporal and/or spatial coordinates), and, say, neural-network model parameters, and the output is a solution value predicted by the neural network model.

For training a model to solve the PDE with potentially no known solution values (see Karniadakis et al. (2021)), one can consider a set of training inputs $\{(\phi_i, y_i)\}_{i \in S_1}$ and minimize the average PDE residual over the training inputs. Assuming that, in addition, one has access to observed and/or computed solution data in the form of tuples $\{(\phi_i, y_i, u_i)\}_{i \in S_2}$, one can also aim to minimize the differences between known and predicted solution values. Mathematically, these aims can be expressed as finding $w$ to minimize

$$\frac{1}{|S_1|} \sum_{i \in S_1} \|\mathcal{F}(\phi_i, \mathcal{G}(\phi_i, y_i, w))\|_2^2 \text{ and/or } \frac{1}{|S_2|} \sum_{i \in S_2} \|u_i - \mathcal{G}(\phi_i, y_i, w)\|_2^2. \qquad (4)$$

Note that the $\phi_i$ and/or $y_i$ elements in $\{(\phi_i, y_i)\}_{i \in S_1}$ may be the same or different from those in $\{(\phi_i, y_i, u_i)\}_{i \in S_2}$. Additional terms may also be used for training, e.g., pertaining to initial and/or boundary conditions, or pertaining to *partial* physics information. For example, in §4.2, we train a model for which it is known that a mass-balance equation should hold, so our training problem involves residuals for the known mass-balance equation, even though this only defines the physics partially. Overall, if one combines all learning aims into a single objective function—say, with a linear combination involving weights for the different objective terms—then one is employing a *soft-constrained* approach to learning. We contend that a more effective approach can be to take at least a subset of terms and impose them as *hard constraints* during training. For example, with respect to the aims in (4), one might impose constraints such as $\mathcal{F}(\phi_i, \mathcal{G}(\phi_i, y_i, w)) = 0$ for some $i \in S_1$ and/or $u_i = \mathcal{G}(\phi_i, y_i, w)$ for some $i \in S_2$. Our experiments show the benefits of this idea.

## 4 EXPERIMENTS

In this section, we present the results of numerical experiments that compare the performance of our proposed methodology (`P-Adam(con)`, where "con" stands for "constrained") versus a soft-constrained approach with Adam scaling (`Adam(unc)` for "unconstrained") (Karniadakis et al., 2021) and a hard-constrained approach with (*projection-less*) Adam scaling (`Adam(con)`) (Márquez-Neila et al., 2017). We consider four test problems. A few of them—namely, our 1D spring, 1D Burgers' equation, and 2D Darcy flow problems—have been seen in the literature; see Li et al. (2021); Négiar et al. (2023). We also consider a problem from chemical engineering, a modified version of a reaction network proposed in Gupta et al. (2016). To ensure a fair comparison, for a test problem, `P-Adam(con)`, `Adam(unc)` and `Adam(con)` use the same objective function, which is usually the data-fitting loss plus PDE residual, while the hard-constrained methods `P-Adam(con)` and `Adam(con)` impose additional constraints: the PDE residuals are zero at some input data points. Further details are provided in each problem's subsection. A GitHub repository containing the implementations of each of the algorithms and our test problems can be found at **[to be inserted in non-anonymized version]**. The software uses PyTorch (BSD-3 license). For all algorithms and all experiments, the Adam parameters $\beta_1 = 0.9$, $\beta_2 = 0.999$, and $\mu = 10^{-7}$ were used; see Algorithm 2. Our numerical experiments were performed using Google CoLaboratory™ V100 GPU platforms. We estimate that it would require about one week to reproduce all of our experimental results. We discuss a comparison of the running times of the three methods in Appendix C.

### 4.1 1D SPRING

Our first test problem aims to predict the movement of a damped harmonic (mass-spring) oscillator (Moseley, 2018) under the influence of a restoring force and friction. For simplicity, our aim was

to train a model to predict the movement of the spring for known parameters and a single initial condition. (Our later test problems involved more complicated situations; this simple problem and the case of only a single initial conditions merely serves as a good starting point for comparison.) The spring can be described by a linear, homogeneous, second-order ordinary differential equation with constant coefficients, namely, $m\frac{d^2u(t)}{dt^2} + \mu\frac{du(t)}{dt} + ku(t) = 0$ over $t \in [0,1]$, where we fixed the mass $m = 1$, friction coefficient $\mu = 4$, and spring constant $k = 400$. This corresponds to an under-damped state for which the exact solution with amplitude $A$ and phase $\phi$ is well known to be $u(t) = e^{-\delta t}(2A\cos(\phi + (\sqrt{w_0^2 - \delta^2})t))$, where $\delta = \mu/(2m)$ and $w_0 = \sqrt{k/m}$.

Our aim was to train a neural network with the known ODE and a few observed solution values to be able to predict the height of the spring at any time $t \in [0,1]$. We used a fully connected neural network with 1 input neuron (corresponding to $t$), 3 hidden layers with 32 neurons each, and 1 output neuron (that predicts the spring height at time $t$). Hyperbolic tangent activation is used at each hidden layer. For the training problems, we used two types of terms: ODE-residual and data-fitting terms. The times at which the ODE-residual terms were defined were 30 evenly spaced points over $[0,1]$. The times at which the data-fitting terms were defined were 10 evenly spaced points over $[0,0.4]$. The runs for `Adam(unc)` only considered an objective function where the terms in (4) were combined with a weight of $10^{-4}$ on the average ODE-residual. The runs for `Adam(con)` and `P-Adam(con)` considered the same objective and included hard constraints for the ODE residual at times $\{\frac{4}{29}, \frac{12}{29}, \frac{21}{29}\}$, i.e., 3 constraints. For all algorithms, we ran a "full-batch" version (i.e., with exact objective gradients employed) and a "mini-batch" version, where in each iteration of the latter version only half of the ODE-residual data points were used. For consistency in the experiments, rather than employ the step-size rule in Algorithm 1, we employed the same two fixed learning rates (i.e., value of $\alpha_k$ for all $k \in \mathbb{N}$) for each algorithm: $5 \times 10^{-4}$ and $1 \times 10^{-4}$.

Results are provided in Figures 1 and 2. The plots in Figure 1 show that `P-Adam(con)` yielded lower objective values (loss) more quickly and achieved better accuracy (i.e., lower mean-squared error for the objective terms) after the training budget expired. They also show that `P-Adam(con)` achieved more comparable results for the two learning rates, whereas the other algorithms performed worse for the smaller learning rate. Results for a wider range of learning rate tuning can be found in Figure 11a in Appendix B. Appendix A also compares the robustness of the three methods on smaller and larger neural network sizes. Our results here demonstrate that `P-Adam(con)` requires less hyperparameter tuning. The plots in Figure 2 show that the difference in performance can be seen clearly in the predictions that one obtains. Appendix F compares the ODE residuals of the three methods and demonstrates that `P-Adam(con)` achieves the smallest ODE residuals across the entire time window. Appendix E provides additional results for varying hard constraints.

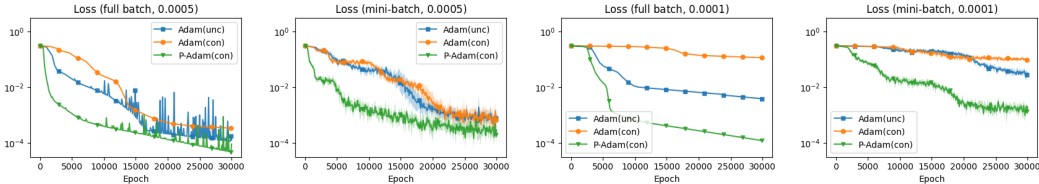

Figure 1: 1D Spring losses over epochs. For the mini-batch runs, the solid lines indicate means over 5 runs while the shaded regions indicate values within one standard deviation of the means.

## 4.2 CHEMICAL ENGINEERING PROBLEM

This problem models the reaction system of 1-butene isomerization when cracked on an acidic zeolite (Gupta et al., 2016). The system is reformulated as an ordinary linear differential equation by scaling the kinetic parameters of the true model so that the equations are more flexible. The ODE is

$$\frac{du(t)}{dt} = \begin{bmatrix} -(c^{(1)} + c^{(2)} + c^{(4)})u^{(1)}(t) + c^{(3)}u^{(3)}(t) + c^{(5)}u^{(4)}(t) \\ 2c^{(1)}u^{(1)}(t) \\ c^{(2)}u^{(1)}(t) - c^{(3)}u^{(3)}(t) \\ c^{(4)}u^{(1)}(t) - c^{(5)}u^{(4)}(t) \end{bmatrix},$$

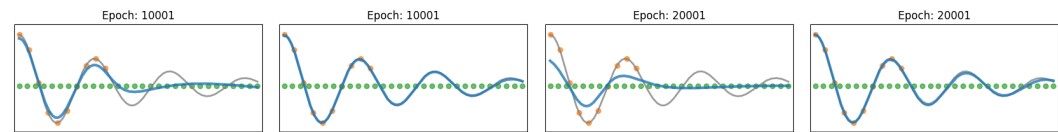

Figure 2: Predicted trajectories. Left to right: `Adam(unc)` (mini-batch, $\alpha_k = 0.0005$), `P-Adam(con)` (mini-batch, 0.0005), `Adam(unc)` (mini-batch, 0.0001), and `P-Adam(con)` (mini-batch, 0.0001). Axes are time $t \in [0,1]$ (horizontal) and true/predicted $u(t)$ (vertical). Green dots indicate times at which the ODE-residual terms are defined; orange dots indicate data-fitting values; the gray line indicates the true solution; and the blue line indicates the predicted solution. Code from Moseley (2018) (available under the MIT License) is used to generate the plots.

where $c = [4.283, 1.191, 5.743, 10.219, 1.535]^T$. Our aim was to train a neural network with the known ODE and mass-balance condition (namely, $\frac{du^{(1)}(t)}{dt} + 0.5\frac{du^{(2)}(t)}{dt} + \frac{du^{(3)}(t)}{dt} + \frac{du^{(4)}(t)}{dt} = 0$) over various initial conditions near a nominal initial condition, where the nominal one is $u_0 = [14.5467, 16.335, 25.947, 23.525]$. In this manner, the trained network can be used to predict $u(t)$ at any $t$ (we use the range $t \in [0, 10]$) for any initial condition near the nominal one.

We used a fully connected neural network with 5 input neurons (corresponding to initial condition in $\mathbb{R}^4$ and $t \in \mathbb{R}$), 3 hidden layers with 64 neurons each and hyperbolic tangent activation, and 4 output neurons (corresponding to $u(t) \in \mathbb{R}^4$). The training problems involved three objective terms: ODE-residual (weighted by $10^{-2}$), mass-balance (weighted by $10^{-2}$), and data-fitting (weighted by 1) terms. Training data was generated by solving the ODE over 1000 initial conditions (of the form $u_0 + \xi$, where $\xi$ was a random vector with each element drawn from a uniform distribution over $[-1, 1]$) using `odeint` from the `scipy` library (Virtanen et al., 2020) (BSD licensed). Specifically, solution values were obtained over 64 evenly spaced times in $[0, 10]$, which over the 1000 initial conditions led to 64000 training points. The ODE-residual and mass-balance terms involved all 64000 training points, whereas the data-fitting term involved only 20% of these points chosen at random with equal probability. The runs for `Adam(unc)` used only these objective terms, whereas the runs for `Adam(con)` and `P-Adam(con)` considered in addition to 10 constraints on mass-balance residuals, the points for which were chosen uniformly at random over all initial conditions and times. The mini-batch size was 20% of all samples. We tested learning rates for all algorithms: $5 \times 10^{-4}$ and $1 \times 10^{-4}$. The results in Figures 3 and 4 show that `P-Adam(con)` performed best.

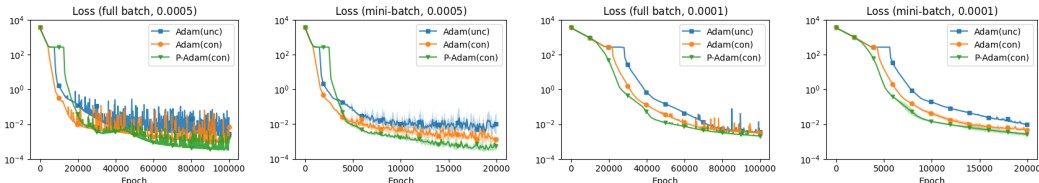

Figure 3: Chem. eng. problem losses over epochs. For the mini-batch runs, solid lines indicate means over 5 runs while the shaded regions indicate values within one standard deviation of the means.

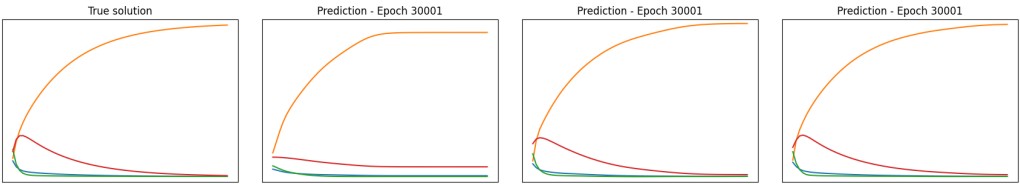

Figure 4: True/predicted solutions for the setting (full batch, $\alpha_k = 0.0001$) after 30000 epochs for an initial condition not in the training set. Left to right: true solution (first), prediction by `Adam(unc)` (second), prediction by `Adam(con)` (third), and prediction by `P-Adam(con)` (fourth). Inspection reveals that the `P-Adam(con)` prediction is closer to the true solution even at this stage in training.

## 4.3 1D BURGERS' EQUATION

Burgers' equation is a partial differential equation often used to describe the behavior of certain types of nonlinear waves (Wang et al., 2021; Négiar et al., 2023). With respect to a spatial domain $[0, 1]$, time domain $[0, 1]$, and viscosity parameter $\nu = 0.01$, we used the equation, initial condition, and (periodic) boundary condition

$$\frac{\partial u(x,t)}{\partial t} + u(x,t)\frac{\partial u(x,t)}{\partial x} = \nu\frac{\partial^2 u(x,t)}{\partial x^2}, \qquad x \in (0,1),\ t \in [0,1];$$
$$u(x,0) = u_0(x), \qquad x \in [0,1];$$
$$u(x,t) = u(x+1,t), \qquad x \in [0,1],\ t \in [0,1].$$

Our aim was to train a neural network with the known PDE and boundary condition over various initial conditions near a nominal initial condition. In this manner, for any $(x, t)$ and initial condition near the nominal one, the trained network can predict $u(x, t)$.

We used a fully-connected neural network with 34 input neurons (corresponding to $x$, $t$, and a discretization of $u_0$ over 32 evenly spaced points), 3 hidden layers with 64 neurons each and hyperbolic tangent activation, and 1 output neuron (corresponding to $u(x, t)$). The training problems involve three objective terms: PDE-residual (weighted by $10^{-3}$), boundary-residual (weighted by $10^{-3}$), and data-fitting (weighted by 1) terms. Training data was generated by solving the PDE over 100 initial conditions (of the form $u_0(x) = \sin(2\pi x + \xi\pi)$, where for each instance $\xi$ was chosen from a uniform distribution over $[0, 0.2]$) using the `odeint` solver, as in the previous section. Specifically, solution values were obtained over 32 evenly spaced points each in the spatial and time domains, which over the 100 initial conditions led to 102,400 training points. For each initial condition, the PDE-residual and boundary-residual terms involved all relevant generated training points, whereas the data-fitting term involved only 200 points chosen at random with equal probability. The `Adam(unc)` used only these objective terms, whereas `Adam(con)` and `P-Adam(con)` considered in addition to 10 constraints on PDE residuals, the points for which were chosen uniformly at random over all initial conditions and spatio-temporal points. The mini-batch was 20% of all samples. We tested learning rates: $10^{-3}$ and $5 \times 10^{-4}$. One finds in Figure 5 that the results obtained by `Adam(unc)` and `P-Adam(con)` were in fact comparable, although the performance by the projection-less Adam approach (`Adam(con)`) was inferior. Figure 6 shows that a prediction by the model obtained by `P-Adam(con)` is indeed close to the true solution. Additional numerical results obtained when tuning the learning rate over a wider range along with different neural network sizes can be found in 11b in Appendix B and Appendix A, respectively. The main take-away from these results is the superior performance of `P-Adam(con)` over `Adam(con)`.

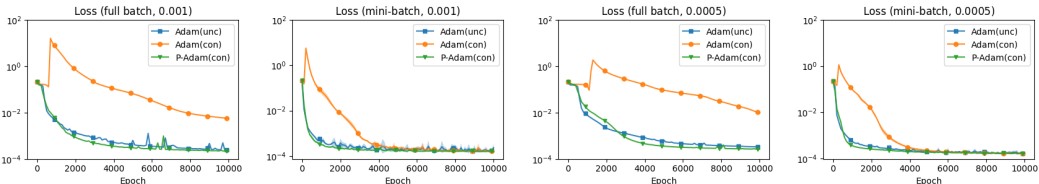

Figure 5: Burgers' losses over epochs. For mini-batch runs, solid lines indicate means over 5 runs while the shaded regions (not very visible) indicate values within one standard deviation of the means.

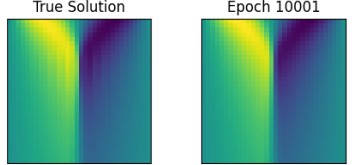

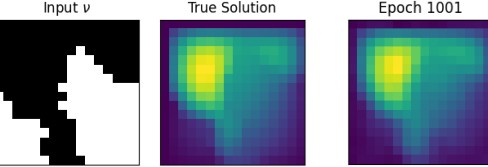

Figure 6: Burgers' true/predicted solutions for initial condition not seen in training. Predicted solution by `P-Adam(con)` (mini-batch, $\alpha_k = 0.0005$).

Figure 7: Darcy flow diffusion coefficient $\nu$ and true/predicted solution, where diffusion coefficient $\nu$ not seen in training. Predicted solution by `P-Adam(con)` (mini-batch, $\alpha_k = 0.005$).

## 4.4 2D DARCY FLOW

The steady-state 2D Darcy flow equations model the flow of a fluid through a porous medium (Négiar et al., 2023; Takamoto et al., 2023). With respect to the spatial domain $[0,1]^2$, a forcing function $f$, and a diffusion coefficient $\nu$, we used

$$-\nabla \cdot (\nu(x)\nabla u(x)) = f(x), \ x \in (0,1)^2; \quad u(x) = 0, \ x \in \partial[0,1]^2.$$

Our aim was to train a neural network with the known PDE and boundary condition over various diffusion coefficients such that, for any $x \in [0,1]^2$ and $\nu$, it could be used to predict $u(x)$.

We used the Fourier Neural Operator (FNO) architecture imported from the *neuralop* library (Kovachki et al., 2023; Li et al., 2021) (MIT License). The inputs were given in three channels, one for $\nu$, one for a horizontal position embedding and one for a vertical position embedding. Each channel had dimension $16 \times 16$. We used 4 hidden layers (the default). The output was a single channel of dimension $16 \times 16$ (corresponding to $u(x)$). The training problems involve three objective terms: PDE-residual (weighted by $10^{-2}$), boundary-residual (weighted by $10^{-2}$), and data-fitting (weighted by 1) terms. Training data was generated by solving the PDE over 1000 $\nu$ values, the values and corresponding solutions of which were obtained by *neuralop* using default settings. For each $\nu$ value, the PDE-residual and boundary-residual terms involved all relevant generated training points, whereas the data-fitting term involved only 20% of the points chosen at random with equal probability. The runs for `Adam(unc)` used only these objective terms, whereas the runs for `Adam(con)` and `P-Adam(con)` considered the same objective in addition to 50 constraints on PDE residuals, the points for which were chosen uniformly at random over all initial conditions and spatial points. We ran full-batch and mini-batch settings, where the mini-batch was dictated by 20% of the $\nu$ values. We tested using the same learning rates for all algorithms: $5 \times 10^{-3}$ and $5 \times 10^{-4}$. For these problems, plots of losses make it harder to distinguish between the algorithms. Therefore, in Figure 8, we plot PDE-residual loss values only, the results of which show preferable performance by `P-Adam(con)` over the other methods. Figure 7 shows that a prediction by the model obtained by `P-Adam(con)` is close to the true solution.

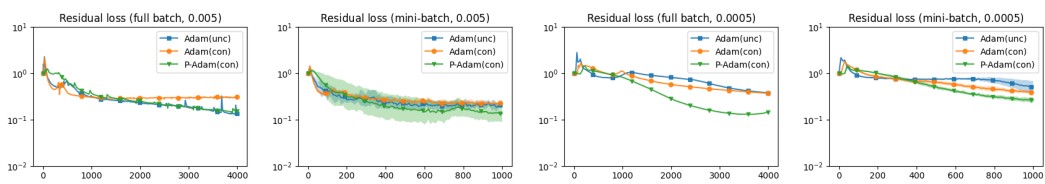

Figure 8: Darcy flow losses over epochs. For mini-batch runs, solid lines indicate means over 5 runs while the shaded regions indicate values within one standard deviation of the means.

## 5 CONCLUSION AND FUTURE WORK

We proposed a method for informed learning that is stochastic-gradient-based, handles hard constraints, and employs a novel projection-based Adam diagonal scaling. The method's per-iteration cost is comparable to an unconstrained (soft-constrained) approach that also uses diagonal scaling. Numerical experiments reveal practical benefits of the proposed scheme, which we conjecture would also be witnessed when training larger and more sophisticated neural networks for informed learning.

Future work includes proving that the theoretical convergence guarantees of the stochastic SQP method extend when our `P-Adam(con)` strategy is used. The analysis could be based on convergence guarantees of the Adam method that have been established in recent years (e.g., in Zhang et al. (2022)). The recent two stepsize SQP method in O'Neill (2024) is also a way to accumulate the momentum of a step only in the null space of the constraint Jacobian. Comparisons of the performance of that method with `P-Adam(con)` would also be relevant for future work.

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

## A  EXPERIMENTS WITH DIFFERENT NEURAL NETWORK DEPTHS AND WIDTHS

To show that our proposed method (`P-Adam(con)`) is easier to tune over different neural network sizes, we conducted experiments on the 1D spring and 1D Burgers' equation problems to compare the obtained losses over different neural network depths and widths. The results are shown in Figures 9 and 10. The results show that, among the three methods, `P-Adam(con)` is usually the most robust one to different neural network depths and widths, and always achieves the best performance.

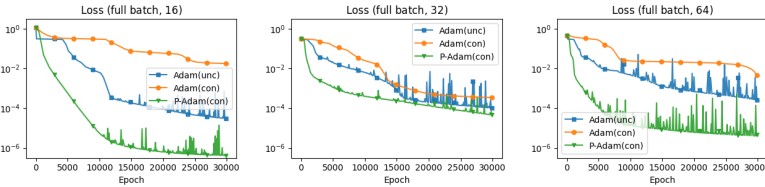

(a) 1D Spring losses over 16, 32, and 64 neurons each hidden layer. The remaining settings are the same as in Figure 1 (full batch, learning rate $5 \times 10^{-4}$).

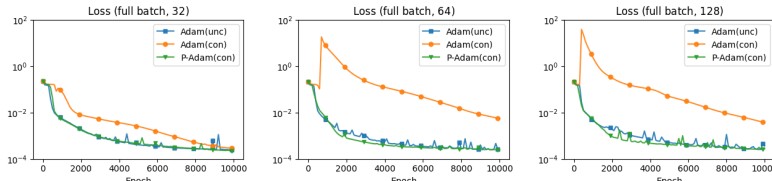

(b) Burgers' losses over 32, 64, and 128 neurons each hidden layer. The remaining settings are the same as in Figure 5 (full batch, learning rate $10^{-3}$).

Figure 9: Losses over different neural network widths.

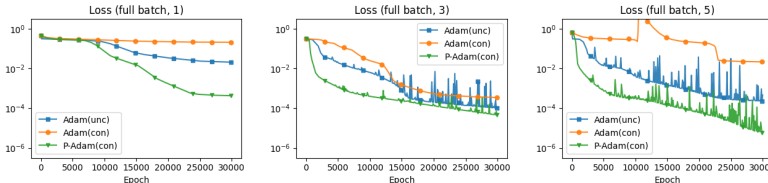

(a) 1D Spring losses over 1, 3, and 5 hidden layers. The remaining settings are the same as in Figure 1 (full batch, learning rate $5 \times 10^{-4}$)

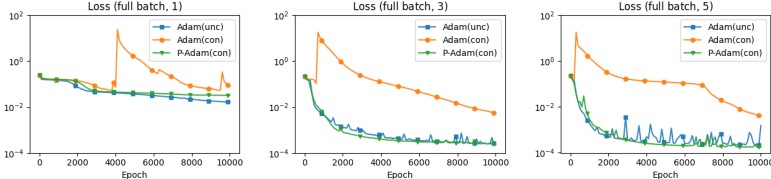

(b) Burgers' losses over 1, 3, and 5 hidden layers. The remaining settings are the same as in Figure 5 (full batch, learning rate $10^{-3}$).

Figure 10: Losses over different neural network depths.

## B  EXPERIMENTS WITH DIFFERENT LEARNING RATES

In this section, we exhibit the robustness of our method to the learning rate through experiments on the 1D spring and 1D Burgers' equation problems. We tested learning rates in a wide range of $\{10^{-2}, 5 \times 10^{-3}, 10^{-3}, 5 \times 10^{-4}, 10^{-4}\}$ for these two problems. The results are shown in Figure 11. Similar to the robustness to the neural network size as shown in Appendix A, the results here demonstrate the impressive robustness of `P-Adam(con)` with respect to the learning rate.

Moreover, `P-Adam(con)` usually converges faster than other two methods across all five learning rates. In contrast, `Adam(con)` even fails to converge with the largest learning rate $10^{-2}$ when applied to the Spring problem.

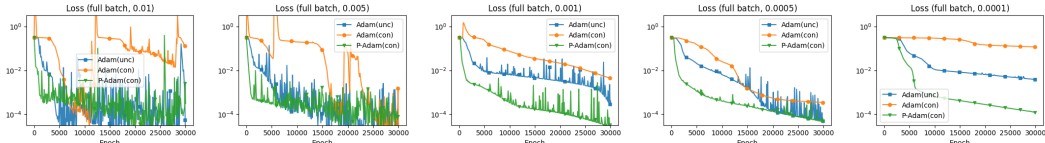

(a) 1D Spring losses over learning rates: $10^{-2}, 5 \times 10^{-3}, 10^{-3}, 5 \times 10^{-4}, 10^{-4}$. The remaining settings are the same as in Figure 1 (full batch).

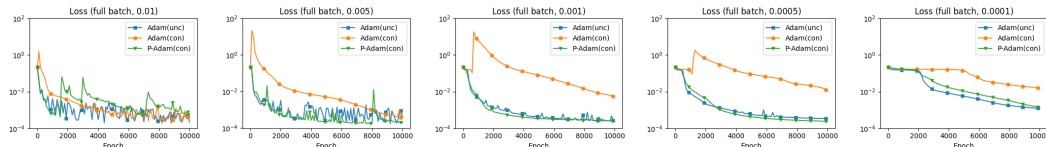

(b) Burgers' losses over learning rates: $10^{-2}, 5 \times 10^{-3}, 10^{-3}, 5 \times 10^{-4}, 10^{-4}$. The remaining settings are the same as in Figure 5 (full batch).

Figure 11: Losses over different learning rates.

## C   RUNNING TIME

We ran the three methods for all problems with full batch and learning rate $5 \times 10^{-4}$ for a fixed number of epochs (iterations) on a Colab L4 GPU python 3 node. Running time statistics are shown in the Table 1.

We have several observations from the table. First, `Adam(con)` and `P-Adam(con)` exhibit similar running times. This is expected since the primary difference between the two methods is that `P-Adam(con)` requires projecting the stochastic gradient onto $\text{Range}(\nabla c(w_k))$, which is not more computationally expensive since both methods need to compute the projection of $c(w_k)$ onto $\text{Range}(\nabla c(w_k))$ when computing the $v$ component of the step. Second, for problems with either 3 or 10 constraints, the running time of `Adam(con)` and `P-Adam(con)` is approximately double that of `Adam(unc)`. For the problem with 50 constraints, the running time is roughly three times that of `Adam(unc)`. This is also expected, as `Adam(con)` and `P-Adam(con)` need to solve two linear systems when computing the $v$ and $u$ steps, with the size of the linear system matrix being the number of constraints by the number of trainable parameters. However, it is worth noting that the computation time for `Adam(con)` and `P-Adam(con)` could be improved by using a more efficient linear system solver, such as `minres`. Currently, we are using `torch.linalg.solve(A, b)` from PyTorch. While PyTorch does not support the `minres` solver, other libraries, such as SciPy, do. Nevertheless, improving the efficiency of solving linear systems is not our focus in this work, so we opted for the straightforward approach available in PyTorch. Still, even though our proposed method has a higher per-iteration cost, one can see through all of the experiments in the paper and these appendices that, in many cases, our proposed method yields a better trained model if one were to have a computational time budget.

Table 1: Running time(s) per iteration

|  | 1D Spring | Chem. eng. | 1D Burgers | 2D Darcy flow |
|---|---|---|---|---|
| Running time (s) Adam(unc) | 0.009 | 0.024 | 0.042 | 0.852 |
| Running time (s) Adam(con) | 0.019 | 0.055 | 0.059 | 2.568 |
| Running time (s) P-Adam(con) | 0.020 | 0.053 | 0.060 | 2.533 |
| # constraints for Adan(con) and P-Adan(con) | 3 | 10 | 10 | 50 |
| # trainable parameters | 2209 | 8964 | 10625 | 22101 |

## D    PERFORMANCE OF ALGORITHM 1

In our experiments, Algorithm 1 converges more slowly than Algorithm 2, which utilizes the momentum of the projected gradient. Figure 12 shows the performance of the 1D Spring problem in the mini-batch setting, where Algorithm 1 is represented as `SGD(unc)` or `SGD(con)`, depending on the presence of hard constraints. Notably, the difference is evident even in the unconstrained setting, as seen when comparing `Adam(unc)` with `SGD(unc)`. This observation motivated us to develop momentum methods.

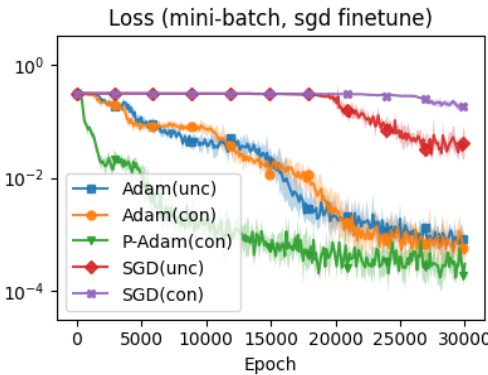

Figure 12: Comparison of the performance of Algorithm 1 and Algorithm 2 on the 1D Spring problem. The learning rate for `Adam(unc)`, `Adam(con)`, and `P-Adam(con)` are $5 \times 10^{-4}$ while for `SGD(unc)` and `SGD(con)` are $10^{-2}$ (after tuned).

## E    EFFECTS ON DIFFERENT NUMBER OF HARD CONSTRAINTS

One may wonder about the effects of changing the number of hard constraints in our method. For example, when we use a selection of ODE/PDE residuals as hard constraints, we can define a hard constraint for each ODE/PDE input by using the residual at that input. In this case, the number of hard constraints equals the number of selected inputs. Alternatively, we can aggregate the residuals from multiple inputs to define a single hard constraint.

In this section, we test the performance of our method when varying the number of hard constraints and when define hard constraints by aggregating over multiple inputs. As an example, we use the 1D Spring problem in the mini-batch setting with a step size of $0.0005$. The results are shown in Figure 13.

Our results show that: (1) Increasing the number of hard constraints by selecting more inputs improves the training loss performance, as seen when comparing the green and purple curves. However, this comes at the cost of increased computational effort.(2) Using the same number of hard constraints but aggregating the residuals over more inputs does not significantly change the computational cost. It may lead to faster convergence, but the training loss after convergence may not improve, as observed when comparing the green and red curves.

## F    SPRING PROBLEM ODE RESIDUAL VISUALIZATION

We present Figure 14 and Figure 15 to visualize the ODE residual over the time window $[0, 1]$ for the 1D Spring problem discussed in Section 4.1. The ODE residual here is defined as $m\frac{d^2u(t)}{dt^2} + \mu\frac{du(t)}{dt} + ku(t)$. Figure 14 shows the distribution of residuals over five random runs at each discrete time, while Figure 15 illustrates the average absolute residuals. The results demonstrate the following: (1) our method `P-Adam(con)` achieves the smallest ODE residual across all discretized times. As shown in Figure 14, the boxes in the third plot are closest to zero at all times compared to the first and second plots. (2) The `Adam(unc)`, as a soft-constrained method, exhibits larger ODE residuals

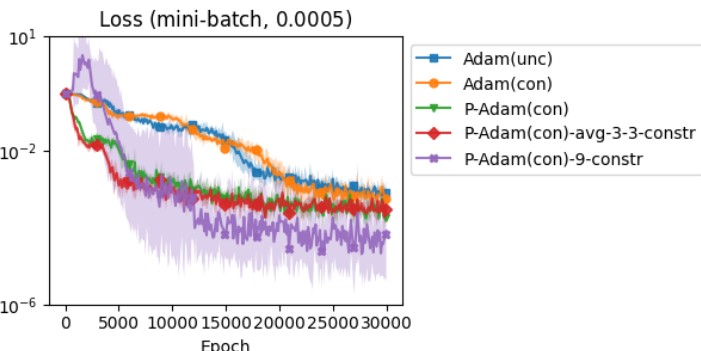

Figure 13: Training loss comparison for varying hard constraints. `Adam(unc)`, `Adam(con)`, and `P-Adam(con)` correspond to the settings in Section 4.1 and Figure 1, using three hard constraints. Both `P-Adam(con)-avg-3-3-constr` and `P-Adam(con)-9-constr` use input times at $3, 4, 6, 7, 12, 15, 20, 21, 22/29$. `P-Adam(con)-avg-3-3-constr` averages residuals over three consecutive times and as a result uses three hard constraints, while `P-Adam(con)-9-constr` uses nine hard constraints at the specified times. For 30000 epochs, `P-Adam(con)` runs in 1,230 seconds, `P-Adam(con)-avg-3-3-constr` in 1,300 seconds, and `P-Adam(con)-9-constr` in 2,300 seconds.

than the hard-constrained methods. (3) The ODE residuals are significantly reduced at and near the times treated as hard constraints, i.e., $\{\frac{4}{29}, \frac{12}{29}, \frac{21}{29}\}$, when comparing the soft-constrained method (`Adam(unc)`) to the hard-constrained methods.

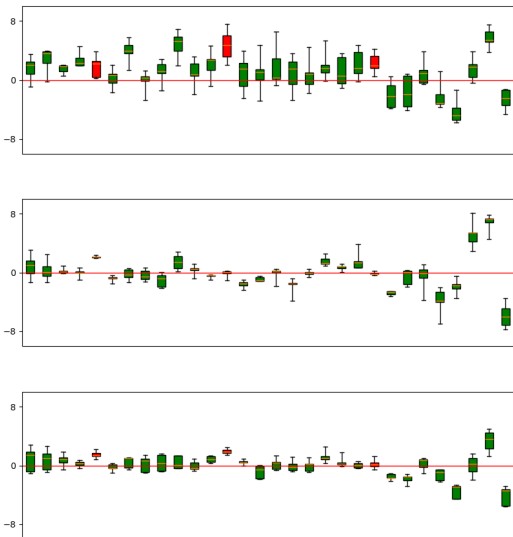

Figure 14: Spring problem ODE residuals over time window $[0, 1]$. From top to bottom, the plots correspond to `Adam(unc)`, `Adam(con)`, and `P-Adam(con)`, respectively. All results are based on the (mini-batch, step size $= 0.0005$) setting. Each box represents the terminated ODE residual over 5 random runs at the corresponding time. Green boxes indicate times treated as soft constraints, while red boxes correspond to times treated as hard constraints, specifically $\{\frac{4}{29}, \frac{12}{29}, \frac{21}{29}\}$ as described in Section 4.1.

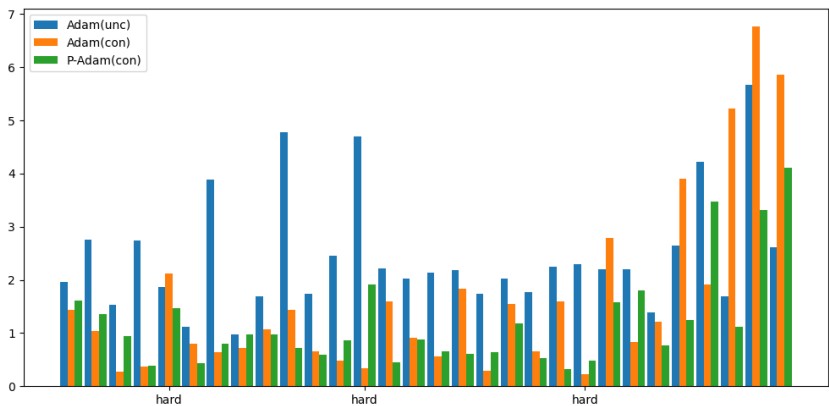

Figure 15: Spring problem average ODE residuals of 5 random runs over time window $[0, 1]$. The experiment setting is the same as for Figure 14.

