# OpenReview forum: "Informed Machine Learning with a Stochastic-Gradient-based Algorithm for Training with Hard Constraints"
_ICLR.cc/2025/Conference — Submitted to ICLR 2025_

### Official Review · Reviewer_nK6C · 2024-10-24

**Soundness:** 3
**Presentation:** 2
**Contribution:** 2
**Rating:** 5
**Confidence:** 3

**Summary:**

The paper brings together the stochastic gradient-based SQP framework by Berthas et al. and the popular Adam optimization algorithm for ML-based solving of partial/ordinary differentiable equations with hard constraints.

**Strengths:**

The topic of the study is interesting.
I am happy to revise my evaluation based on the answers to my questions.

**Weaknesses:**

One may argue that novelty and originality are limited.
Algorithm 1 and the underlying Theorem 1 are based on previous work (Berthas et al., 2021).
Except the empirical evaluations, the main novel aspect boils down to plugging in an Adam-like update rule into Algorithm 1.

The basic idea in Algorithm 2 that the „stochastic gradient gk is replaced by its orthogonal projection onto the null space of ∇c(wk )T“ in a mixture of Adam and Algorithm 1 is clear. However, this key aspect - perhaps almost trivial - should be spelled out because it is so important. That is, line 1 in Algorithm 2 should be derived as used in the algorithm with all intermediate steps.

The manuscript states „When the number of rows of J (i.e., m) is small, the overall cost is proportional to that of computing H−1g with a symmetric and positive definite H, as is required for an Adam-based method for the unconstrained setting.“ But there a no non-diagonal matrices in place in Adam, right? Is this under the assumption that H is diagonal (e.g. the identity matrix I)?  Why is the scaling of Adam and P-Adam then the same?

In fact, the scaling w.r.t. to the number of constraints could be problematic.
Although wall clock time experiments are always a bit problematic (e.g., the may depend on the solver), I suggest to plot performance over wall clock time depending on the number of hard constraints.
The number of hard constraints in the experience was always very small, some scaling experiments would be interesting to see.

## Experiments

The differences between the methods are not always very pronounced.
The are only small differences in general in the „Chemical Engineering Problem“ and between Adam(inc) and P-Adam in 1D Burgers’ Equation, and also in general on the „2D Darcy flow“ task.

Baseline experiments using  the stochastic gradient-based SQP with steepest descent as in Algorithm 1 are missing.
How much do we actually gain from moving to P-Adam if we tune the SGD learning rate?

The reader wonders how well the „hard“ constraints are met.  For example, what is the value of the mass balance during the course of optimisation for the different methods?
It would be nice to see the ODE-residual errors for the points that were treated as soft targets vs this which were linked to „hard“ constraints.

1D Spring: The times at which the ODE-residual terms were defined were 30 evenly spaced points over [0,1] (end page 6).
The P-Adam considered the hard constraints on ODE-residuals at the time points   {0.14, 0.4, 0.7}.
These were different points that the equally spaced 30, right?
If yes: Is this fair? Why was no subset selected? How do the results change if for the settings without hard constraints the three points are added to the soft constraints (i.e., 33 instead of 30 points are computed in the ODE-residual  error component of the error)?

## Comments regarding presentation

The listing of the methods in lines 297-299 is confusing. Should it be „Adam(con)“ instead of „Adam(unc)“ in line 299?

Rather put Appendix C on runtime in the main body of the paper and move some specification of the (standard) benchmark problems to the appendix.

**Questions:**

- How is line 1 in Algorithm 2 exactly derived? Could you spell this out so that it is easy to follow (it is one of the main aspects of the study)?
- Why is it correct to state (lines 240-243) that computing H^-1g is proportional to what is require in a standard Adam step?
- How much do we actually gain from moving from  stochastic gradient-based SQP as in Algorithm 1 to P-Adam if we tune the SGD learning rate?
- How is the wall-clock time scaling w.r.t to the number of "hard" constraints?
- How well are the "hard" constraints met in practice? Nor always exactly, right?

---

> ### Author Response · Authors · 2024-11-18
>
> **Weakness**:
> One may argue that novelty and originality are limited. Algorithm 1 and the underlying Theorem 1 are based on previous work (Berthas et al., 2021).
> Except the empirical evaluations, the main novel aspect boils down to plugging in an Adam-like update rule into Algorithm 1.
>
> **Response**: Although Algorithm 1 and its corresponding Theorem 1 are based on previous work (Berthas et al., 2021), it is novel to employ it for solving physics-informed machine learning problems using the hard-constrained problem formulation.  Otherwise, indeed, a main contribution of our work is the incorporation of an Adam-type update rule.  Our experiments show that doing this in a direct manner, such as in Márquez-Neila et al. (2017), does not lead to consistently better results.  However, the use of our projection-based Adam technique performs much better, and often outperforms a soft-constrained approach.
>
> **Weakness**:
> The basic idea in Algorithm 2 that the ``stochastic gradient $g_k$ is replaced by its orthogonal projection onto the null space of $\nabla c(w_k)T$'' in a mixture of Adam and Algorithm 1 is clear. However, this key aspect - perhaps almost trivial - should be spelled out because it is so important. That is, line 1 in Algorithm 2 should be derived as used in the algorithm with all intermediate steps.
>
> **Response**: We thank the reviewer for this comment. We are happy to change the presentation of Algorithm 2 and include the entire loop of the algorithm as in Algorithm 1. We will update the manuscript PDF on November 21.
>
> **Weakness**:
> The manuscript states ``When the number of rows of $J$ (i.e., $m$) is small, the overall cost is proportional to that of computing $H^{-1}g$ with a symmetric and positive definite $H$, as is required for an Adam-based method for the unconstrained setting.'' But there a no non-diagonal matrices in place in Adam, right? Is this under the assumption that $H$ is diagonal (e.g. the identity matrix I)? Why is the scaling of Adam and P-Adam then the same?
>
> **Response**: This appears in Line 242. Yes, it is under the assumption that $H$ is diagonal, as our statement in Line 113-115. We will edit the manuscript to add the word ``diagonal" in Line 242 and upload on November 21. The details of the computational cost can be found in our response to Reviewer FQAx, i.e., the overall cost of Algorithm 2 is $\mathcal{O}(m^2n + m^3)$. When $m \ll n$, it is $\mathcal{O}(n)$, which is the same as the computation cost of Adam.
>
> **Weakness**:
> In fact, the scaling w.r.t. to the number of constraints could be problematic. Although wall clock time experiments are always a bit problematic (e.g., the may depend on the solver), I suggest to plot performance over wall clock time depending on the number of hard constraints. The number of hard constraints in the experience was always very small, some scaling experiments would be interesting to see.
>
> **Response**: As in our response to Reviewer FQAx, we will add experiments comparing performance when increasing the number of hard constraints by using more samples in defining hard constraints. In addition, as our response to Reviewer CkE3, one way to use more samples to compose hard constraints without increasing the total number of constraints (to account for computational costs) is that one can generate certain clusters of samples, and for each cluster, the average PDE residual can be treated as a hard constraint to be zero. We are happy to incorporate such experiments to make a comparison and will update the manuscript PDF with these details by November 21.

---

> > ### Author Response · Authors · 2024-11-18
> >
> > (authors response continued)
> >
> > **Experiment comment 2**:
> > Baseline experiments using the stochastic gradient-based SQP with steepest descent as in Algorithm 1 are missing. How much do we actually gain from moving to P-Adam if we tune the SGD learning rate?
> >
> > **Response**: We are happy to incorporate experiments to compare Algorithm 1 (i.e., not using momentum stochastic gradients) with P-Adam. We will update the manuscript PDF with these results by November 21.
> >
> > **Experiment comment 3**:
> > The reader wonders how well the **hard** constraints are met. For example, what is the value of the mass balance during the course of optimisation for the different methods? It would be nice to see the ODE-residual errors for the points that were treated as soft targets vs this which were linked to ``hard'' constraints.
> >
> > **Response**: We are happy to incorporate results that demonstrate the ODE/PDE-residual errors for points treated as soft constraints and hard constraints. We will update the manuscript PDF with these results by November 21.
> >
> > **Experiment comment 4**:
> > 1D Spring: The times at which the ODE-residual terms were defined were 30 evenly spaced points over [0,1] (end page 6). The P-Adam considered the hard constraints on ODE-residuals at the time points {0.14, 0.4, 0.7}. These were different points that the equally spaced 30, right? If yes: Is this fair? Why was no subset selected? How do the results change if for the settings without hard constraints the three points are added to the soft constraints (i.e., 33 instead of 30 points are computed in the ODE-residual error component of the error)?
> >
> > **Response**: We thank for this question. Yes, the points $\\{0.14, 0.4, 0.7\\}$ used for hard constraints might not belong to the set for defining soft-constraints in the objective. We will adjust the experiment setting so that the hard constraints are defined over a subset of samples defining the soft constraints.  (Generally speaking it does not need to be a subset, but we are happy to make this change for our comparison here.)  We will update the manuscript PDF with these results by November 21.
> >
> > **Comments regarding presentation 1**:
> > The listing of the methods in lines 297-299 is confusing. Should it be Adam(con) instead of Adam(unc) in line 299?
> >
> > **Response**: Yes. Thanks for catching this typo. We will fix it in the PDF.
> >
> > **Comments regarding presentation 2**:
> > Rather put Appendix C on runtime in the main body of the paper and move some specification of the (standard) benchmark problems to the appendix.
> >
> > **Response**: Thanks for the suggestion. We will move the running time in the main body and adjust other parts of the presentation accordingly to ensure the page limit is satisfied.
> >
> > **Question 1**:
> > How is line 1 in Algorithm 2 exactly derived? Could you spell this out so that it is easy to follow (it is one of the main aspects of the study)?
> >
> > **Response**: Line 1 in Algorithm 2 takes the projection of the stochastic gradient $g_k$ onto the null space of $\nabla c(w_k)$. Note the matrix $\nabla c(w_k)(\nabla c(w_k)^T\nabla c(w_k))^{-1}\nabla c(w_k)^T$ is the projector of the range space of $\nabla c(w_k)$ and $I-\nabla c(w_k)(\nabla c(w_k)^T\nabla c(w_k))^{-1}\nabla c(w_k)^T$ is a projector of null space of $\nabla c(x_k)^T$. Also, $$I-\nabla c(w_k)(\nabla c(w_k)^T\nabla c(w_k))^{-1}\nabla c(w_k)^T = Z_k(Z_k^TZ_k)^{-1}Z_k^T$$ where the columns of $Z_k$ span the null space of $\nabla c(w_k)^T$. We use the project matrix $I-\nabla c(w_k)(\nabla c(w_k)^T\nabla c(w_k))^{-1}\nabla c(w_k)^T$ since it is computable through the algorithm. Our intuition of taking this projection is presented in Line 181-199.
> >
> > **Question 2-5** are addressed through the responses above.

---

> > > ### Author Response · Authors · 2024-11-23
> > >
> > > Dear Reviewer, we have updated the PDF. The changes mentioned to be updated on November 21 have been incorporated. Appendix F discusses the performance of increasing hard constraints by selecting more inputs or using the same number of constraints by aggregating ODE residuals at multiple times. Appendix F also shows, in the Spring problem, how the PDE residual (both soft and hard constraints) behaves after training.
> > >
> > > For **Experiment Comment 4**, we found that the experiment indeed uses a subset of soft constraints to define hard constraints. Specifically, we selected times from the soft constraints that are closest to {0.14, 0.4, 0.7} to define the hard constraints. This resulted in times {4/29, 12/29, 21/29}. We have corrected this in the PDF.
> > >
> > > For **Comments Regarding Presentation 2**, we did not move Appendix C to the main body in this version, as we have added other appendix sections that may also be considered for inclusion in the main body. We might consider doing so in future versions.

---

> > > > ### Comment · Reviewer_nK6C · 2024-11-25
> > > >
> > > > I have read the rebuttal(s).
> > > > The efforts to improve the manuscript are appreciated.
> > > >
> > > > One of my main questions/concerns was about how ODE-residual errors for the points that were treated as soft targets compared to points which were linked to ``hard'' constraints.
> > > > To address this, the authors added Appendix F / Figure 14. This is appreciated. However, I would have preferred to see a plot like this for all problems. More important, I think the plot does not look very convincing. One might argue that Adam(con) did a better than P-Adam(con) for the hard constraints. Furthermore, the differences in residuals between soft and "hard" constraints are not very pronounced. I think this needs to be studied further.

---

> > > > > ### Author Response · Authors · 2024-11-27
> > > > >
> > > > > We thank the reviewer for their further comments.
> > > > > Figure 14 plots the box plot of $c(w_T)$ of five random runs where $w_T$ represents the terminated solutions. To facilitate an easier comparison of residuals across the three algorithms, we added Figure 15 to the PDE. Figure 15 shows the average $|c(w_T)|$ of five random runs for all three algorithms in a grouped bar chart. Over thirty discrete times, P-Adam(con) achieves the smallest residual at 16 times, Adam(con) 13 times, and Adam(unc) only once.
> > > > >
> > > > > - We can prepare plots similar plots to Figure 14 for the other three problems in the coming days and include them in the camera-ready version. Over our experiments, P-Adam consistently gives the best average ODE/PDE-residuals over all inputs that define soft constraints (i.e., a superset of the points defining the hard constraints) among the three methods. Therefore, P-Adam(con) would demonstrate the smallest residuals at most inputs, as observed in the Spring problem.
> > > > >
> > > > > - Regarding the concern about comparing the performance of Adam(con) and P-Adam(con) at hard constraints residuals, we emphasize that both algorithms aim to find a KKT solution of the constrained problem (1) in our manuscript , i.e., a point $w$ associated with some $\lambda$ such that
> > > > > \begin{align*}
> > > > > \nabla f(w) + \nabla c(w) \lambda = 0 \quad \text{ and }\quad c(w) = 0.
> > > > > \end{align*}
> > > > > Therefore, a solution $w$ with better constraint residual $||c(w)||$ does not necessarily indicate a better solution overall. For the PIML problem, Figures 1, 3, and 5 display the total training loss, which combines some data-fitting loss and some residual of the differential equations. We believe this metric is a more meaningful measure of algorithm performance in solving PIML problems. Additionally, Figures 2, 4, 6, and 7 demonstrate that P-Adam(con) produces accurate predictions, suggesting that the predictions adhere to the physical rules governing the system.
> > > > >
> > > > > - For the concern that differences in residuals between soft and hard constraints are not very pronounced, we respectfully disagree. We believe the differences are indeed significant. We highlight that ODE/PDE constraints exhibit a strong dependency on neighboring inputs. For instance, in the Spring problem, the first hard constraint is:
> > > > > \begin{equation*}
> > > > >     m \frac{d^2 u(t)}{d t^2} + \mu \frac{d u(t)}{d t} + k u(t) = 0 \quad \text{at}\quad t = \frac{4}{29}.
> > > > > \end{equation*}
> > > > > If this quantity is close to zero, we expect the constraint value to also be close to zero at, e.g., $t=\frac{3}{29}, \frac{5}{29}$, etc. This behavior is evident in Figure 14, as we discussed in Line 863: "The ODE residuals are significantly reduced at and near the times treated as hard constraints, i.e., $\{\tfrac{4}{29}, \tfrac{12}{29}, \tfrac{21}{29}\}$, when comparing the soft-constrained method (Adam(unc)) to the hard-constrained methods."
> > > > >
> > > > >     We would also point out to the reviewer that the prediction function family that is chosen has an effect on the ultimate residual that can be obtained.  For example, for a given neural network architecture, there are limits to how accurately the PDE can be satisfied at all inputs.  Our work does not focus on neural network design for solving physics-informed problems; that is the subject of the researchers of others in the field.  Rather, our work aims to show that---whatever the architecture/model that is chosen---our strategy can offer better algorithmic behavior and final solution quality compared to alternative methods.

---

### Official Review · Reviewer_FQAx · 2024-11-02

**Soundness:** 3
**Presentation:** 3
**Contribution:** 3
**Rating:** 6
**Confidence:** 2

**Summary:**

This paper investigates informed machine learning by proposing a novel approach that integrates hard constraints directly into the optimization process, as opposed to previous methods that formulate informed constraints as soft constraints relying on penalty techniques (e.g., augmented Lagrangian methods). Building upon a recent stochastic gradient descent (SGD) method for constrained optimization, the authors incorporate hard constraints using a Newton-based technique. By employing a projection-based approach, they enable the handling of hard constraints within the well-established Adam optimizer. The method is empirically demonstrated in several small-scale experiments to exhibit robust and superior performance compared to other methods that do not treat constraints as hard constraints.

**Strengths:**

- **Hard constraint handling in Adam** The paper proposes to integrate projected gradient descent with Lagrange multipliers into the stochastic optimization framework, specifically adapting the Adam optimizer to handle hard constraints. By modifying the standard Adam update rule to project gradients onto the null space of the constraint Jacobian, ensures current solution feasibility at each iteration.
- **Computational Efficiency in small constraint scale**: When the Hessian matrix is diagonally approximated and has a small constraint number (low-rank constraint Jacobian), the constraint-related equations can be solved efficiently which does not trigger much computationally overhead.

**Weaknesses:**

- **Scalability of constraints**: While the authors argue that a small number of constraints often suffices for good performance, the scalability of the method as the number of constraints increases is not extensively discussed. As the number of constraints grows, the projection step may become less efficient, and solving the constraint-related equations can become computationally intensive. It would be beneficial for the authors to elaborate, either theoretically or empirically, on the algorithm's limitations and performance when dealing with a larger set of constraints, this could be accomplished on more scalable problems, which also help tackle the limited empirical scope weakness mentioned below.

- **Lack of Convergence proof** Although the paper references the convergence properties of the SGD-SQP method, these theoretical guarantees do not naturally extend to the proposed P-Adam algorithm while left as future work.

- **Limited Empirical Scope** The experimental validation presented in the paper is limited to small-scale problems with relatively simple neural network models. This restricted scope makes it challenging to assess the method's effectiveness and scalability in more complex or large-scale applications. Expanding the empirical evaluation to include larger datasets and more sophisticated models would strengthen the paper's claims regarding the practical utility of the proposed method.

**Questions:**

- How is the method different and similar to reference [1], which deals with linear equality constraint in PINN through projection layers derived from KKT conditions

[1] https://arxiv.org/abs/2402.07251

---

> ### Author Response · Authors · 2024-11-18
>
> **Weakness 1**: **Scalability of constraints**: While the authors argue that a small number of constraints often suffices for good performance, the scalability of the method as the number of constraints increases is not extensively discussed. As the number of constraints grows, the projection step may become less efficient, and solving the constraint-related equations can become computationally intensive. It would be beneficial for the authors to elaborate, either theoretically or empirically, on the algorithm's limitations and performance when dealing with a larger set of constraints, this could be accomplished on more scalable problems, which also help tackle the limited empirical scope weakness mentioned below.
>
> **Response**: We thank the reviewer for this comment. For the computational cost, we denote the number of constraints as $m$. The main computation cost of Algorithm 2 is Line 1 and Line 6. In Line 1, the computation cost of computing $\bar{g}_k$ is as follows: cost of $(\nabla c(w_k)^T\nabla c(w_k))^{-1}$ is $\mathcal{O}(m^2n + m^3)$, cost of $(\nabla c(w_k)^T\nabla c(w_k))^{-1}\nabla c(w_k)^Tg_k$ is $\mathcal{O}(m^2n + m^3 + m^2 + mn)$. Hence the cost of computing $\bar{g}_k$ is $\mathcal{O}(m^2n + m^3)$. For Line 6 when computing $s_k$, as discussed in Section 2.3, we discuss the cost of computing $u$ and $v$ for $s = u + v$. The cost of computing $v$ (Line 225) is $\mathcal{O}(m^2n + m^3)$. The cost of computing $u$ (Line 238) is $\mathcal{O}(m^2n + m^3)$, similarly to computing $\bar{g}_k$. Therefore, the overall of Algorithm 2 is $\mathcal{O}(m^2n + m^3)$. When $m \ll n$, it is $\mathcal{O}(n)$.
>
> Empirically, we can add experiments comparing performance when increasing the number of hard constraints by using more samples in defining the hard constraints. In addition, as in our response to Reviewer CkE3, one way to use more samples to compose hard constraints without increasing the total number of constraints (to account for computational costs) is that one can generate certain clusters of samples, and for each cluster, the average PDE residual can be treated as a hard constraint to be zero. We are happy to incorporate such experiments to make a comparison. We will update the manuscript PDF with these details by November 21.
>
> **Question**: How is the method different and similar to reference (https://arxiv.org/abs/2402.07251), which deals with linear equality constraint in PINN through projection layers derived from KKT conditions
>
> **Response**: We thank the reviewer for this question and bringing this reference. As our response to Reviewer CkE3, the method in Chen et al. (https://arxiv.org/abs/2402.07251) is limited to constraints of the form $Bu(x) + Ax = b$, where $x$ is the input, $u(x)$ is the neural network approximated PDE solution, and $(A, B, b)$ are given parameters. In other words, their method is applicable only to linear relationships between PDE inputs and solutions. Consequently, their approach cannot be applied to solve all the problems in our experiments, where the constraints involve nonlinear relationships between the input and the PDE solution. In contrast, our method is capable of handling problems with general nonlinear and nonconvex constraints. Additionally, their use of projection differs from ours: they project the model predictions onto the feasible region defined by the linear equality constraints, whereas we use the projected stochastic gradient only when taking momentum.  Otherwise, our approach allows iterates to be infeasible, which is the case for the best-performing state-of-the-art methods for solving constrained optimization problems.

---

> > ### Author Response · Authors · 2024-11-23
> >
> > Dear reviewer, we have updated the PDF. The changes mentioned will be updated on November 21 are incorporated. Appendix F discussed the performance for increading hard constraints by selecting more inputs or using the same number of constraints by aggregating ODE residuals at multiple times.

---

> > > ### Comment · Reviewer_FQAx · 2024-11-26
> > >
> > > Thank you to the authors for the detailed response regarding the theoretical complexity analysis and the empirical comparison of constraint numbers in Appendix E, Figure 13. I have two additional concerns:
> > >
> > > - Could the authors explain the possible cause of the initial loss bumps observed in the 9-constraint cases?
> > > - Upon further review of Figure 14, I share a similar question with reviewer nK6C: could the authors clarify why the error performance at the hard constraint times is similar to that at other times? Additionally, it appears that the error at constraint times for Adam(con) is quite competitive with that of P-Adam(con).

---

> > > > ### Author Response · Authors · 2024-11-27
> > > >
> > > > We thank the reviewer for further engagement and questions.
> > > >  - **For question 1**, the addition of more hard constraints has two effects: It adds more nonlinearity to the problem and restricts further the search directions in early iterations.  This initially has the effect of steering the algorithm toward solutions with larger PDE residuals, as shown in the table below at Epochs 100, 1000, and 5000. However, as the algorithm progresses, the 9-hard-constraint model yields smaller overall PDE residuals. This is reasonable since the additional constraints helps to guide the algorithm to better solutions, ultimately leading to improved predictions and a lower total loss. We are happy to add this discussion to Appendix E.
> > > >
> > > >     **Table**: performance of P-Adam(con) and P-Adam(con)-9-constr at selected epochs. Both are the running performance at one of the five randome runs. \|\|c\|\|_inf represents the $\ell\infty$ norm of the hard constraints.
> > > >
> > > >     |       | P-Adam(con) |          |                   |                 |   | P-Adam(con)-9-constr |          |                   |                 |
> > > >     |-------|-------------|----------|-------------------|-----------------|---|----------------------|----------|-------------------|-----------------|
> > > >     | Epoch | Loss        | PDE Residual | Data-fitting Loss | \|\|c\|\|_inf |   | Loss                 | PDE Residual | Data-fitting Loss | \|\|c\|\|_inf |
> > > >     | 0     | 3.10E-01    | 1.46E+01 | 3.09E-01          | 4.14E+00        |   | 3.10E-01             | 1.46E+01 | 3.09E-01          | 4.17E+00        |
> > > >     | 100   | 3.08E-01    | 1.83E+01 | 3.06E-01          | 3.64E+00        |   | 3.10E-01             | 1.15E+01 | 3.09E-01          | 3.65E+00        |
> > > >     | 1000  | 7.79E-02    | 4.78E+02 | 3.01E-02          | 1.19E+01        |   | 3.72E+01             | 3.42E+05 | 2.93E+00          | 7.36E+02        |
> > > >     | 5000  | 5.86E-03    | 5.28E+01 | 5.89E-04          | 3.74E+00        |   | 2.80E-01             | 1.46E+03 | 1.34E-01          | 1.41E+01        |
> > > >     | 10000 | 3.39E-03    | 3.34E+01 | 4.68E-05          | 3.13E+00        |   | 8.15E-05             | 7.70E-01 | 4.49E-06          | 3.10E+00        |
> > > >     | 20000 | 7.28E-04    | 6.26E+00 | 1.03E-04          | 1.71E+00        |   | 3.68E-05             | 3.50E-01 | 1.82E-06          | 1.63E+00        |
> > > >     | 30000 | 2.71E-04    | 2.49E+00 | 2.17E-05          | 1.43E+00        |   | 2.02E-04             | 1.99E+00 | 3.51E-06          | 1.55E+00        |
> > > > * **For question 2**,
> > > >     1. For the concern that the error performance at the hard constraints is similar to that at other times, we highlight that ODE/PDE constraints exhibit a strong dependency on neighboring inputs. For instance, in the Spring problem, the first hard constraint is:
> > > > \begin{equation*}
> > > >     m \frac{d^2 u(t)}{d t^2} + \mu \frac{d u(t)}{d t} + k u(t) = 0 \quad \text{at}\quad t = \frac{4}{29}.
> > > > \end{equation*}
> > > > If this quantity is close to zero, we expect the constraint value to also be close to zero at, e.g., $t=\frac{3}{29}, \frac{5}{29}$, etc. This behavior is evident in Figure 14, as we discussed in Line 863: "The ODE residuals are significantly reduced at and near the times treated as hard constraints, i.e., $\{\tfrac{4}{29}, \tfrac{12}{29}, \tfrac{21}{29}\}$, when comparing the soft-constrained method (Adam(unc)) to the hard-constrained methods."
> > > >
> > > >     2. Regarding the concern about comparing the performance of Adam(con) and P-Adam(con) at hard constraints residuals, we emphasize that both algorithms aim to find a KKT solution of the constrained problem (1), i.e., a point $w$ associated with some $\lambda$ such that
> > > >     \begin{align*}
> > > >     \nabla f(w) + \nabla c(w) \lambda = 0 \quad \text{ and }\quad c(w) = 0.
> > > >     \end{align*}
> > > >     Therefore, a solution $w$ with better constraint residual $||c(w)||$ does not necessarily indicate a better solution overall. For the PIML problem, Figures 1, 3, and 5 display the total training loss, which combines some data-fitting loss and some residual of the differential equations. We believe this metric is a more meaningful measure of algorithm performance in solving PIML problems. Additionally, Figures 2, 4, 6, and 7 demonstrate that P-Adam(con) produces accurate predictions, suggesting that the predictions adhere to the physical rules governing the system.

---

### Official Review · Reviewer_29rd · 2024-11-03

**Soundness:** 2
**Presentation:** 2
**Contribution:** 2
**Rating:** 5
**Confidence:** 4

**Summary:**

This paper introduces an approach that characterized by three key aspects. Initially, it incorporates prior information into the training process via hard constraints instead of the more common contemporary technique of soft constraints. Furthermore, the approach abstains from using penalty-based methods. Instead, it relies on a recently introduced stochastic-gradient-based algorithm that is computationally efficient and employs a Newton-based method for constraint management. Lastly, a projection-based adaptation of the widely recognized Adam optimization algorithm is suggested for scenarios involving hard constraints. The numerical experiments achieve superior final prediction accuracy when contrasted with a soft-constraint method.

**Strengths:**

This article proposes a potential new method and demonstrates good results in numerical experiments.

**Weaknesses:**

* The narrative in the first section of this paper is somewhat disorganized; it does not clearly articulate the motivation behind the paper, which is, why it is essential to employ key techniques such as hard constraints, and why penalty-related algorithms are not utilized. Since the number of hyperparameters for soft constraints is not significantly large, the core difference between soft and hard constraints is not clearly identified.

* The serious issue with this article is that it exaggerates their contributions. The contributions stated in section 1.1 (a), (b), and the entire content of section 2.1 are in fact all derived from [1]; the authors have not made any form of innovation. The only innovative part of the entire article is section 2.2.

* The innovation in section 2.2 is also quite confusing. Algorithm 2 is very similar to the algorithm in [2], with the only difference being the application of a projection operator to the gradient. However, the article's explanation of why the projection is used is confusing. The conclusion in the article is that when H is chosen as the identity matrix I, the gradient and the projected gradient are the same, but the problem is that this clearly does not hold for the Adam algorithm. The article does not provide any other explanation for this distinction, nor does it have convergence theory to support it. The experimental results alone are not convincing.



[1] Berahas A S, Curtis F E, Robinson D, et al. Sequential quadratic optimization for nonlinear equality constrained stochastic optimization[J]. SIAM Journal on Optimization, 2021, 31(2): 1352-1379.

[2] Márquez-Neila P, Salzmann M, Fua P. Imposing hard constraints on deep networks: Promises and limitations[J]. arXiv preprint arXiv:1706.02025, 2017.

**Questions:**

*

---

> ### Author Response · Authors · 2024-11-18
>
> **Response to Weakness 1**: Our main motivation is to develop a hard-constrained method for solving physics-informed machine learning problems. Through our experiments, we demonstrate that our hard-constrained approach (P-Adam-con), which incorporates additional hard constraints, outperforms the penalty-based method (Adam-unc), as discussed in Lines 57–62. Our intuition is that hard constraints guide the neural network to prioritize mapping the PDE solution even only at certain inputs, enabling faster and more efficient training. We are happy to incorporate this explanation into Section 1.  We will upload a revised PDF by November 21.
>
> **Response to Weakness 2**:
> We disagree with the assertion that Sections 1.1(a) and 1.1(b) exaggerate our contributions. Regarding Section 1.1(a), our proposed method---Algorithm 1 with steps computed using Algorithm 2---is indeed a novel approach for solving general constrained stochastic objective optimization problems.  (To say this is not novel is to say that Adagrad and Adam are not distinct from each other, which we do not think is a reasonable assertation.) This innovation extends naturally to solving hard-constrained physics-informed machine learning problems. For Section 1.1(b), the computational efficiency of our method is addressed in Section 2.3, where we detail an efficient approach for solving the linear system (3) when the matrix $H$ is diagonal. This contribution is also novel and highlights a key advantage of our approach.
>
> Regarding Section 2.1, we did not claim to have designed Algorithm 1 or developed Theorem 1. These were appropriately cited from Berahas et al. (2021), (in Line 109-110, Line 136, Line 162-165). Instead, the purpose of Section 2.1 is to introduce this algorithmic framework and demonstrate how it can be applied to solve physics-informed machine learning problems with hard constraints. This framework forms the foundation of our new proposed method, Algorithm 2, which features a unique step computation mechanism.
>
> We are happy to incorporate these clarifications into the manuscript.  We will upload a revised PDF by November 21.
>
> **Response to Weakness 3**: The difference between our method and that of Márquez-Neila et al. (2017), namely, the application of a projection operator to the gradient, is a crucial modification that results in superior performance, as demonstrated in the experiments section.  (In many cases, the hard-constrained approach of Márquez-Neila et al. (2017) offers worse performance than a soft-constrained approach, whereas with our modification the hard-constrained approach indeed becomes better than a soft-constrained one.)  The rationale for employing this projection is explained in Lines 181–191. When $H$ is the identity, the component of the search direction in the range space of $\nabla c(w_k)$, denoted by $v_k$, can be computed by $v_k = -\nabla c(w_k) (\nabla c(w_k)^T \nabla c(w_k))^{-1} c(w_k)$, and the component in the null space of  $\nabla c(w_k)^T$, denoted by $u_k$, can be computed as $u_k = -Z_k(Z_k^TZ_k)^{-1}Z_k^Tg_k$. Note $Z_k(Z_k^TZ_k)^{-1}Z_k$ is a projection operator onto $\text{Null}(\nabla c(w_k)^T)$. In other words, the search direction can be decomposed into two parts: one part is independent of the current evaluated stochastic gradient $g_k$, while the other part is solely the projected stochastic gradient. Consequently, when taking momentum of the stochastic gradient to compute the search direction, the components of the stochastic gradients lying in $\text{Range}(\nabla c(w_k))$---which do not affect the current step computation---should not be accumulated. Therefore, we project out this component, as shown in Line 1 of Algorithm 2.
>
> The reviewer should also recognize that our explanation is applicable even though the Adam approach does not use $H$ as the identity matrix.  It should be understood this way: Adam involves running averages of gradient and squared-gradient components.  In other words, it does not involve running averages of scaled gradient and squared-gradient components.  For the same reason, our projected Adam method involves running averages of projected gradient and projected squared-gradient components, even though the ultimate step is computed with the diagonal scaling applied.  Overall, our approach is entirely consistent with Adam for the unconstrained setting.

---

> > ### Author Response · Authors · 2024-11-23
> >
> > Dear reviewer, we have updated the PDF. The changes mentioned will be updated on November 21 are incorporated.

---

> > ### Comment · Reviewer_29rd · 2024-11-25
> >
> > I have re-read the revised article, and I appreciate the authors' addition of a distinction description for contributions compared to previous work, as well as the inclusion of more experiments. However, my main concern still remains: why is the application of the projection operator more effective? The statement in Section 2.2 of the paper mainly indicates that this change is feasible, but why is it better? This is indeed an unusual change, so you need more description to demonstrate its effectiveness, especially for basic optimization algorithms, where high performance in experiments alone is not enough. Therefore, I believe the article still needs further improvement. However, given the authors' detailed rebuttal, I will raise the score.

---

### Official Review · Reviewer_CkE3 · 2024-11-04

**Soundness:** 3
**Presentation:** 3
**Contribution:** 2
**Rating:** 5
**Confidence:** 3

**Summary:**

The paper proposes a new methodology in the area of physics-informed machine learning by incorporating “hard constraints” during stochastic-gradient-based training. The approach differs from traditional “soft-constrained” methods that add penalty terms to the objective function, which can be difficult to tune and less effective. The main innovation is a novel projection-based variant of the Adam optimizer (P-Adam), adapted for hard-constrained optimization. Numerical experiments show that this approach outperforms traditional Adam on four tasks: a 1D spring oscillator, a chemical engineering reaction model, 1D Burgers' equation, and 2D Darcy flow.

**Strengths:**

Strengths
- Unlike conventional methods based on penalty terms, the use of hard constraints directly embeds domain-specific knowledge. The performance improvements based on this are demonstrated in the case studies.
- The authors include a rigorous discussion of the optimization method, starting from the original constrained Sequential Quadratic Programming (SQP) framework settings.
- The experimental results reveal that the proposed methodology leads to better prediction accuracy and requires fewer hyperparameter adjustments, which is advantageous for real-world applications where tuning may be computationally expensive.

**Weaknesses:**

Weaknesses
- The novelty of this work could use clarification. The new techniques do not seem to differ methodologically from the related algorithms presented, e.g., in  Márquez-Neila et al., Berahas et al., and Curtis et al., except the introduction of momentum methods. Moreover, projection of optimization steps given hard constraints has been demonstrated, e.g., by Chen et al. (https://arxiv.org/abs/2402.07251).
- The experimental settings considered have questionable relevance to the state of the art in this area. The methods are only compared against standard and soft-constrained Adam, rather than the bulk of methods for physics-informed machine learning. Moreover, the generalization to practical problems is unclear. Only small-scale problems are considered, enabling the use of non-stochastic gradient descent, and half the data in the batch setting (which may not be realistic).

**Questions:**

Questions
- Pg 3 suggests that the Lipschitz constants for the objective gradient and constraint Jacobian can be estimated. What effects does this have on the convergence or the exactness of the method?
- Can “almost-surely” be defined precisely throughout? Does this refer to a probability?
- Pg 6 suggests that only a few terms should be treated as “hard constraints.” Is there a systematic way to determine which terms should be treated as hard constraints?

---

> ### Author Response · Authors · 2024-11-18
>
> **Weakness 1**: The novelty of this work could use clarification. The new techniques do not seem to differ methodologically from the related algorithms presented, e.g., in Márquez-Neila et al., Berahas et al., and Curtis et al., except the introduction of momentum methods. Moreover, projection of optimization steps given hard constraints has been demonstrated, e.g., by Chen et al. (https://arxiv.org/abs/2402.07251).
>
> **Response**: Thank you for this comment and pointing out the reference Chen et al., which is related to our work but only thematically.  We are happy to make changes to the manuscript PDF to address this and other comments.  We will upload the revised PDF on November 21.
>
> We will clarify the relationship between Algorithm 1 and the works of Márquez-Neila et al., Berahas et al., and Curtis et al. in the manuscript. Lines 107–110 state that Algorithm 1 is a simplified version of Berahas et al. (2021). On Line 136, immediately following Theorem 1, which presents the convergence theory for Algorithm 1, we also cite Berahas et al. (2021, Corollary 3.14) and Curtis et al. (2023a, Equation (16)).
>
> Algorithm 2 is a newly proposed algorithm introduced in this manuscript. It differs from Berahas et al. (2021, Corollary 3.14) and Curtis et al. (2023a, Equation (16)) in that Algorithm 2 incorporates the momentum of the stochastic gradient. While Algorithm 2 shares some similarities with Márquez-Neila et al. (2017), we clarified the key difference in Lines 182–185. Specifically, Algorithm 2 utilizes the momentum of the component of the stochastic gradient in the null space of $\nabla c(w_k)^T$ rather than the entire stochastic gradient.  This is a nontrivial change.  (For example, the practical diagonal-scaling variants of Adam and Adagrad are considered distinct methods.  Ours is as distinct as these methods are from each other.) In our experiments, this distinction demonstrates the superior performance of our method compared to that of Márquez-Neila et al. (2017).
>
> For the method in Chen et al. (https://arxiv.org/abs/2402.07251), it is limited to constraints of the form $Bu(x) + Ax = b$, where $x$ is the input, $u(x)$ is the neural network approximating the PDE solution, and $(A, B, b)$ are given parameters.  In other words, their method is applicable only to linear relationships between PDE inputs and solutions. Consequently, their approach cannot be applied to solve all the problems in our experiments, where the constraints involve nonlinear relationships between the input and the PDE solution.
>
> Our method is capable of handling problems with general nonlinear and nonconvex constraints. Additionally, their use of projection differs from ours: they project the model predictions onto the feasible region defined by the linear equality constraints, whereas we only use the projected stochastic gradient when taking momentum.  Otherwise, our approach allows iterates to be infeasible, which is the case for the best-performing state-of-the-art methods for solving constrained optimization problems.
>
> **Weakness 2**: The experimental settings considered have questionable relevance to the state of the art in this area. The methods are only compared against standard and soft-constrained Adam, rather than the bulk of methods for physics-informed machine learning. Moreover, the generalization to practical problems is unclear. Only small-scale problems are considered, enabling the use of non-stochastic gradient descent, and half the data in the batch setting (which may not be realistic).
>
> **Response**: Thank you for this comment. However, since it does not specify a particular method for physics-informed machine learning that the reviewer believes is directly comparable to our method, we would prefer some guidance from the reviewer's expertise.  If given guidance, then we would be happy to provide the results of an experiment in the paper.  Other approaches in the literature either fall into the category of soft-constrained methods or are not directly comparable to our approach due to distinct differences in per-iteration cost (e.g., other methods have much higher per-iteration cost) or other features.  We would be happy to include a comparison with a method that the reviewer can direct us to that is directly comparable.

---

> ### Author Response · Authors · 2024-11-18
>
> (authors response continued)
>
> **Question 1**: Pg 3 suggests that the Lipschitz constants for the objective gradient and constraint Jacobian can be estimated. What effects does this have on the convergence or the exactness of the method?
>
> **Response**: For a Lipschitz constant $L \in \mathbb{R}_{>0}$ of an arbitrary function $f$ such that $||f(x) - f(\bar{x})|| \le L||x - \bar{x}||$, any real constant $\bar{L} \ge L$ is also a valid Lipschitz constant for $f$. Hence, theoretically, if larger Lipschitz constants are used for the objective gradient and constraint Jacobian, the step size $\alpha_k$ will be smaller, leading to slower convergence of the iterate sequence generated by Algorithm 1. Conversely, if the estimated Lipschitz constants are smaller than the actual values, the convergence of the iterate sequence is not guaranteed. In the experiments presented in Section 4, we did not estimate the Lipschitz constants. Instead, we followed common practice by tuning the step size. For instance, in the Spring problem, step sizes were selected from $\\{5\times 10^{-4}, 10^{-4}\\}$. Additional tuning experiments and details are provided in Appendix B.
>
>
> **Question 2**: Can “almost-surely” be defined precisely throughout? Does this refer to a probability?
>
> **Response**: In Theorem 1, ``almost-surely'' is defined with respect to all realizations of a run of Algorithm 1, meaning that the probability of the iterate sequence $\\{W_k\\}$ remaining within a convex set $\mathcal{W}$ is 1.  We will incorporate this clarification into the manuscript PDF and upload the revised version on November 21.
>
> **Question 3**: Pg 6 suggests that only a few terms should be treated as “hard constraints.” Is there a systematic way to determine which terms should be treated as hard constraints?
>
> **Response**: There is no universally best method for selecting samples to be treated as hard constraints. Even in soft-constrained approaches, sampling methods are often employed to choose samples that form the penalty term.  This is an active topic of research in the physics-informed learning community. These ideas could be used in hard-constrained methods, too. In this manuscript, our method is to randomly (uniformly) select a predetermined number of samples and treating them as hard constraints. Alternatively, if one wishes to use more samples to compose hard constraints without increasing the total number of constraints (to account for computational costs), other strategies can be employed. For example, one can generate certain clusters of samples, and for each cluster, the average PDE residual can be treated as a hard constraint to be zero.  (In other words, multiple samples can be combined together to form only a single constraint.  This allows complete flexibility in the number of constraints that are added for a given number of samples.)  We are happy to incorporate such experiments to make a comparison and will update the manuscript PDF with these details by November 21.

---

> > ### Author Response · Authors · 2024-11-23
> >
> > Dear reviewer, we have updated the PDF. The changes mentioned will be updated on November 21 are incorporated. Appendix F discussed the performance for increading hard constraints by selecting more inputs or using the same number of constraints by aggregating ODE residuals at multiple times.

---

> > > ### Comment · Reviewer_CkE3 · 2024-11-25
> > >
> > > Thanks for the detailed responses and clarifications. The clarification of mathematical notations (e.g., in Q1-Q2) is helpful and I will increase my score accordingly.

---

### Comment · Area_Chair_odg5 · 2024-11-13
**authors - reviewers discussion open until November 26 at 11:59pm AoE**

Dear authors & reviewers,

The reviews for the paper should be now visible to both authors and reviewers. The discussion is open until November 26 at 11:59pm AoE.

Your AC

---

> ### Comment · Area_Chair_odg5 · 2024-11-25
>
> Dear reviewers,
>
> The authors have provided individual responses to your reviews. Can you acknowledge you have read them, and comment on them as necessary? The discussion will come to a close very soon now:
> - Nov 26: Last day for reviewers to ask questions to authors.
> - Nov 27: Last day for authors to respond to reviewers.
>
> Your AC

---

### Meta-Review · Area_Chair_odg5 · 2024-12-19

**Metareview:**

The paper proposes a methodology for (physics)-informed machine learning based on hard constraints and optimized with a SGD variant. Nearly all reviewers recommended rejection and agreed on the main problems of the paper: not clear what the novelty is given that lots of ML formulations do use hard constraints with an array of optimization algorithms, and much of the paper comes from previous work; limited experiments (lacking important comparison points, small-scale problems).

**Additional Comments On Reviewer Discussion:**

N/A

---

### Decision · Program_Chairs · 2025-01-22

Reject